# LiveXiv - A Multi-Modal Live Benchmark Based on Arxiv Papers Content

**Nimrod Shabtay**[1,2]**, Felipe Maia Polo**[3]**, Sivan Doveh**[2]**, Wei Lin**[4]**, M. Jehanzeb Mirza**[5]**, Leshem Choshen**[6]**,
Mikhail Yurochkin**[6]**, Yuekai Sun**[3]**, Assaf Arbelle**[2]**, Leonid Karlinsky**[6]**, Raja Giryes**[1]
[1] Faculty of Engineering Tel-Aviv University, [2] IBM Research,
[3] Department of Statistics, University of Michigan, USA [4] JKU Linz, Austria, [5] MIT CSAIL. [6] MIT-IBM

## Abstract

The large-scale training of multi-modal models on data scraped from the web has shown outstanding utility in infusing these models with the required world knowledge to perform effectively on multiple downstream tasks. However, one downside of scraping data from the web can be the potential sacrifice of the benchmarks on which the abilities of these models are often evaluated. To safeguard against test data contamination and to *truly* test the abilities of these foundation models we propose LiveXiv: A scalable evolving live benchmark based on scientific ArXiv papers. LiveXiv accesses domain-specific manuscripts at any given timestamp and proposes to automatically generate visual question-answer pairs (VQA). This is done without any human-in-the-loop, using the multi-modal content in the manuscripts, like graphs, charts, and tables. Moreover, we introduce an efficient evaluation approach that estimates the performance of all models on the evolving benchmark using evaluations of only a subset of models. This significantly reduces the overall evaluation cost. We benchmark multiple open and proprietary Large Multi-modal Models (LMMs) on the first version of our benchmark, showing its challenging nature and exposing the models' true abilities, avoiding contamination. Lastly, in our commitment to high quality, we have collected and evaluated a manually verified subset. By comparing its overall results to our automatic annotations, we have found that the performance deviation is indeed minimal ($< 2.5\%$). Our dataset is available online on HuggingFace and our code is available on GitHub.

## 1 Introduction

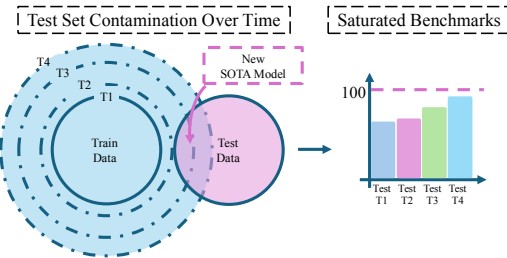

Figure 1: **Static benchmark contamination.** As training data increases, the risk for test set contamination grows and static benchmarks becomes saturated, reflecting falsely improved capabilities.

The internet, with its vast and ever-growing repository of information, serves as a rich data source for training Large Language Models (LLMs) (Brown et al., 2020; OpenAI, 2023; Chiang et al., 2023; Raffel et al., 2019; Touvron et al., 2023a;b; Dubey et al., 2024) and Large Multi-modal Models (LMMs) (OpenAI, 2023; Liu et al., 2023c; Li et al., 2024c; Zhu et al., 2023; Chen et al., 2023a; Alayrac et al., 2022; Radford et al., 2021a). This diverse and continuously updated data fits precisely the need to cover varying knowledge in scale in the training data.

Training on such data enables the models to achieve superhuman performance across a wide range of tasks on multiple common benchmarks (Fu et al., 2023; Yue et al., 2024; Li et al., 2024d; Liu et al., 2023d).

Figure 2: We propose LiveXiv, a new method for generating Live multi-modal dataset for Visual Question-Answering based on ArXiv content. Our pipeline automatically generates scalable and reliable questions along with an efficient evaluation method to reduce the computational and logistic overheads required for continually evaluating past and present models on new versions of the dataset.

> We hypothesize that a portion of LLMs' reported improvements are due to data contamination (Figure 1) and pose the following question: *To what extent does the potential for test set contamination during large-scale training affect our perception of the abilities of LMMs?*

One possible way to safeguard against the contamination of *static* benchmarks is to design a live benchmark that can continuously harness data from the web and turn it into an *ever-evolving* benchmark to test the abilities of these models. A live benchmark may be used in one of the following ways: (a) Expand the dataset over time and evaluate the models' overall knowledge over all collected data, while taking into account that the data might be contaminated. (b) Use only the latest version to assess model capabilities while keeping data contamination risk minimal. While we focus on (b), we share key properties of our efficient evaluation method can be applicable to both cases.

Although, a live benchmark is a promising direction, it still comes with its fair share of challenges. A live benchmark should ideally be updated frequently, consistently, and automatically, *i.e.*, it should be able to scrape the data from the web and formulate it into a benchmark for automated evaluations. Furthermore, as the benchmark is *ever-evolving*, each time a new version arrives, all the participant models need to be re-evaluated, making the update procedure prohibitively expensive both in time and compute. This requires a methodology for efficient evaluation of these models on a continuously updating benchmark. Such a methodology should ease the computational burden of evaluating all the models on each new version of the dataset and reduce the logistic overhead of maintaining inaccessible old models.

In this work, we take a step in this direction and propose LiveXiv – a novel fully automated multi-modal live benchmark that focuses on scientific domains. LiveXiv starts with scraping category-specific (*e.g.*, cs.CV, eess.SY, q-bio.BM, etc.) manuscripts from ArXiv and generates visual question answers from figures, charts, and tables present in these manuscripts through a capable multi-modal model, namely, GPT-4o. As it is challenging to directly feed information-rich PDF documents to GPT-4o, as a pre-processing step, we extract relevant information from the papers by processing it with a structured document parsing pipeline (Team, 2022) to obtain pertinent information like placements of figures, charts, tables, and the text in the captions or in the tables.

This information is used to extract, *e.g.*, by cropping, relevant information from the manuscripts, which is fed to GPT-4o to generate visual questions and answers. Although very capable, GPT-4o is still prone to errors, *e.g.*, due to hallucinations, and may even generate questions that can be answered without visual information. Thus, to mitigate these issues, we add an extensive filtering stage that automatically filters questions requiring only textual information to answer them, and reduce hallucinations through obtaining agreement about the generated questions with another capable multi-modal model, namely, Claude-Sonnet. After the extensive filtering, we obtain a large corpora of VQA pairs which are incorporated into our LiveXiv live benchmark.

Over time, the benchmark is expected to grow, either in the size of the dataset or the amount of models to be evaluated, which increases the required resources for evaluation. Moreover, comparing a new model to existing models at different times requires re-evaluating the existing models over the latest version of the dataset, which can cause additional overhead for continuous evaluation and comparison to prior works. To make the evaluations on LiveXiv feasible, we take inspiration from Maia Polo et al. (2024a;b) and propose a method to approximate the performance of the existing models in new versions of LiveXiv just by re-evaluating a small portion of them. Figure 2 provides a conceptualized overview of our approach.

We summarize our contributions as follows: (a) We propose a scalable live benchmark without any human in the loop that automatically harnesses data from online scientific manuscripts, generates multiple

VQA pairs, filters these questions to reduce errors, and formulates them in the form of a benchmark to test the evolving landscape of LMMs; (b) We introduce an efficient evaluation pipeline that requires LMMs to be tested only on a fraction of the data to infer its performance on the latest version of the benchmark, reducing the overall needed evaluations by at least $\approx 75\%$; (c) We benchmark multiple open and proprietary LMMs on the first version of our benchmark highlighting its challenging nature and providing interesting insights about the models' behavior when evaluated on less contaminated data.

## 2 RELATED WORKS

**Large multi-modal Models (LMMs).** LMMs have shown significant advancements in enabling billion-parameter scale LLMs to perform multi-modal tasks such as image captioning, visual reasoning, and visual question answering. Academia and industry have endeavored to develop LMMs targeting the multi-modal competence of advanced proprietary models like GPT-4o (OpenAI, 2023) and Claude (cla, 2024). Instruct-BLIP performs instruction tuning on the pre-trained BLIP-2 (Li et al., 2023) covering 11 vision-language tasks. The LLaVA series models (Liu et al., 2023c;a;b; Li et al., 2024b) develop the pipeline of collection of instruction-following data and visual instruction tuning with enhanced vision capabilities. The internLM-XComposer (IXC) series (Dong et al., 2024a;b) target free-form vision-language composition and multilingual comprehension. Models from Idefics release (Laurençon et al., 2024b;a) benefit from the massive collection of instruction-following data from over 50 vision-language databases, enhancing capabilities of OCR, document understanding, and visual reasoning. In this work, we include 17 top-performing LMMs in our multi-modal live benchmark LiveXiv, covering both open-sourced and proprietary representatives.

**Static evaluation benchmarks for LMMs.** Most existing LMM benchmarks offer static evaluation with fixed questions and answers (Fu et al., 2023; Yue et al., 2024; Li et al., 2024d; Liu et al., 2023d; Huang et al., 2024; Lin et al., 2024; Zhang et al., 2024b; Onoe et al., 2024; Yanuka et al., 2025). Ben-Kish et al. (2024) use generative models to extend such benchmarks to open settings. MME (Fu et al., 2023) offers evaluation of perception and cognition on 14 tasks and MMMU (Yue et al., 2024) includes 11.5K questions from college exams, quizzes and text books from six major disciplines. Although these benchmarks cover a large variety of multi-modal domain knowledge, evaluation on them is faced with two hazards: the excessive evaluation cost and test data contamination. In this work, we tackle both challenges by proposing a suite that enables efficient evaluation on a contamination-free live benchmark.

**Contamination-free benchmarks.** As large foundation models like LLMs and LMMs are trained on combined sources of tremendous amount of web data or repurposed version of existing open-sourced datasets, there is a high risk of overlap between training data and samples from evaluation benchmarks. Reported evidence and analysis show impact of data contamination on evaluation benchmarks for LLMs (Wei et al., 2023; Zhang et al., 2024a; Cobbe et al., 2021; Roberts et al., 2023; Jain et al., 2024) and LMMs (Chen et al., 2024), indicating the significance of contamination-free evaluation benchmarks. Recent works targeted at LLMs are discussed in A.1.

For LMMs, Vibe-Eval (Padlewski et al., 2024) and LLaVA-Wilder (Li et al., 2024a) perform contamination check on the collected samples that reflect real-world user requests. Most related to our work, the LMMs-Eval LiveBench (Zhang et al., 2024b) collects images from sources of new websites and online forums and employs proprietary LMMs for design and revision of questions. However, the LMMs-Eval LiveBench requires human manual verification of questions which impedes the scalability. Furthermore, it contains only open-ended questions that require LMM-as-a-judge which is time-consuming, susceptible to judge biases, and difficult to scale. In comparison, our LiveXiv constructs a fully-automated data collection pipeline which generates multiple-choice questions which are challenging to the top-performing LMMs.

**Efficient benchmarks.** With the increasing amount of tasks and samples in current benchmarks, evaluation of the full suite is time-consuming and cost-intensive. Efforts are underway to develop efficient benchmarks that reduce computation costs without sacrificing reliability. In A.1 at the appendix we cover the works related to efficient benchmarks.

## 3 LIVEXIV

At a higher level, our automated LiveXiv is created by first obtaining the domain-specific scientific manuscripts from ArXiv at any given timestamp. Then, to obtain pertinent information from the manuscripts, we pass them through a structured document parsing pipeline and then generate visual

Figure 3: Our live dataset generation consists of several stages. We first extract the images and their corresponding metadata (*i.e.* captions and table contents), then we classify the figures into categories using meta-prompting. All the extracted data is then fed to GPT4o to generate multiple questions-answer pairs per image. Since generative models are prone to errors, we apply several filtering steps, using an LLM and LMM to ensure that our dataset is truly multi-modal and reliable.

question answers through a capable LMM (Section 3.1). However, the generated questions can contain errors due to hallucinations or might be too straightforward to answer. Thus, to mitigate these issues, we offer an extensive filtering stage (Section 3.2). To evaluate the benchmark, we propose an efficient evaluation framework to infer the overall performance on the benchmark using only a small subset of evaluations, making the evaluations extremely resource-efficient (Section 3.3). The data acquisition and filtering steps are schematically visualized in Figure 3.

## 3.1 DATA ACQUISITION AND VQA GENERATION

We start with the data acquisition phase, then pre-process the data to obtain the required metadata (*e.g.*, placements of figures, captions, etc.), and then generate the first iteration of VQA from the multi-modal data (figures and tables) from the manuscripts.

**Data Acquisition:** At any given timestamp, we begin by acquiring only ArXiv papers which have non-exclusive license to distribute from predefined domains such as Computer Science (`cs.AI, cs.CV`), Electrical Engineering (`eess.SP, eess.SY`), and Quantitative Biology (`q-bio.BM, q-bio.GN`). However, these manuscripts contain a lot of information that might not be necessary for the task of VQA data generation. Thus, to extract pertinent information we require a pre-processing step.

**Pre-processing:** The downloaded PDFs undergo a structured document parsing pipeline using the DeepSearch toolkit (Team, 2022), which extracts a comprehensive layout of each document, including the positions of figures, tables, captions, and other elements. This structured layout forms the basis for extracting the multi-modal data required for subsequent tasks. To enrich the dataset with additional metadata not captured by the parsing pipeline, we employ a meta-prompting approach with CLIP (Radford et al., 2021b), similar to the method used by Mirza et al. (2024). Specifically, we classify the figures into three distinct categories: Block Diagram, Chart, and Qualitative visual examples which facilitates a more granular, domain-specific evaluation of LMM performance.

**VQA Generation:** For Visual Question Answering (VQA), we construct pairs of figures and their corresponding captions, and for generating VQA from the data present in the tables, we obtain (*e.g.*, crop) images of tables accompanied by their corresponding data.

The VQA process involves two steps using GPT-4o. First, we input the figure and its caption to GPT-4o to generate a detailed description of the figure, employing a Chain-of-Thought (CoT) approach (Wei et al., 2022). Next, the detailed description and figure are fed back into GPT-4o, with prompts adapted from 'ConMe' (Huang et al., 2024) to suit our scientific use case, enabling the generation of relevant VQA questions. For questions from the tables, we utilize the table's content directly, presenting both the image of the table and its data in markdown format to GPT-4o to produce questions that require common-sense reasoning and data manipulation. The automated nature of this process ensures a robust and comprehensive evaluation framework for LMMs, tailored to scientific literature specifics. Detailed prompt templates can be found in Appendix A.6.

## 3.2 FILTERING PHASE

Even though GPT-4o is powerful and has been reported to outperform humans on many different benchmarks (OpenAI, 2023), still it is prone to errors and sometimes can even result in VQA pairs that are answerable without requiring the visual information. Thus, to ensure that the benchmark remains competitive and also has minimum errors, we propose an extensive automatic filtering step. At a higher

level, the filtering phase consists of two main parts, each designed to mitigate a separate issue that can arise due to the automatic dataset generation.

**Blind test with an LLM:** To ensure that the generated VQA pairs are *truly* multi-modal, we pass them through a Large Language Model (LLM), namely LLama-3.1-70B (Meta, 2024), without providing any associated images or image descriptions. This process, referred to as a *blind test*, aims to identify questions that the LLM can answer correctly even in the absence of visual context, indicating they are not truly multi-modal. To ensure robustness, this blind evaluation is repeated multiple times to eliminate any potential *lucky guesses* by the LLM. Questions that are consistently answered correctly by the LLM are filtered out, resulting in the removal of approximately $30\%$ of the generated questions. This step ensures that the remaining questions in the dataset are inherently multi-modal and cannot be answered solely based on linguistic context. The filtered dataset thus represents a more challenging benchmark for evaluating multi-modal capabilities of vision-language models.

**Agreement between disjoint models:** Generative models, including LMMs, are prone to hallucination, where the model generates incorrect or not grounded information. In our case, these hallucinations can lead to erroneous VQA pairs. To address this issue, we introduce an additional filtering step. All questions that pass the initial "blind test" are reviewed along with their generated answers by a different LMM, in this case Claude-Sonnet (cla, 2024), which is provided with the image, question, and the ground truth answer which were all generated by GPT-4o. This second model is asked to either agree or disagree with the generated answer, considering the visual context.

We point out that agreement between models is a nuanced process; incorporating more models to validate answers may lead to the exclusion of difficult questions, thereby diluting the *difficulty* of the dataset. Therefore, we limit this validation step to models with comparable performance to the generation model (*i.e.* GPT-4o). Our preliminary manual evaluation on a subset of the dataset indicates that this agreement step significantly reduces the proportion of incorrect ground-truth (GT) questions, with a reduction of $38.5\%$, while minimally impacting the retention of high-quality question-GT pairs, with only a $6.15\%$ removal of valid pairs. This refinement ensures that the final dataset is both challenging and accurate for the evaluation of LMMs' multi-modal reasoning capabilities. The generated corpus of data is ready to be incorporated into LiveXiv and can be updated automatically without any human intervention.

## 3.3 EFFICIENT EVALUATION

Since LiveXiv is a dynamic benchmark, evaluation can be costly: ideally, whenever a new version of the benchmark is released, all models must be re-evaluated on the updated data, which can pose an engineering challenge and become computationally expensive when handling dozens of models. In this section, we describe our approach to efficient evaluation, which avoids re-evaluating all models at each step, making LiveXiv's maintenance economically feasible. Our idea is based on Item Response Theory (IRT) (Cai et al., 2016; Van der Linden, 2018; Lord et al., 1968; Maia Polo et al., 2024b;a), a collection of statistical models traditionally used in psychometrics and educational assessment. We briefly give some background on IRT and detail how we use it for our evaluations.

### 3.3.1 ITEM RESPONSE THEORY (IRT)

We use the IRT model to predict the probability of a certain LMM $i$ answering correctly on a sample (question) $j$. In mathematical terms, let $Y_{ij} \in \{0,1\}$ denote the correctness on sample $j$ when responded by LMM $i$:

$$Y_{ij} \sim \text{Bernoulli}(\mu(\theta_i, \beta_j)),$$

where $\theta_i$ is an LMM-specific parameter, $\beta_j$ is a sample-specific parameter, and $\mu$ is a function that maps those parameters to the probability of correctness. In this work, we follow Maia Polo et al. (2024b) and assume the parameters live in the real line while $\mu$ induces a logistic regression model. In more detail, we assume

$$\mathbb{P}(Y_{ij}=1;\theta_i,\beta_j) = \frac{1}{1+\exp[-(\theta_i-\beta_j)]}. \tag{1}$$

Here, $\theta_i$ can be interpreted as the skill level of LMM $i$ while $\beta_j$ is seen as the hardness of sample $j$. By equation 1, if $\theta_i$ is much greater (resp. smaller) than $\beta_j$, then the probability $\mathbb{P}(Y_{ij}=1;\theta_i,\beta_j)$ will be close to one (resp. zero). This version of the IRT model is known as the Rasch model (Georg, 1960; Chen et al.,

2023b), and it is widely used in fields such as recommendation systems (Starke et al., 2017), educational testing (Clements et al., 2008), and evaluation of language models (Maia Polo et al., 2024b). Moreover, it has a similar formulation to the popular Bradley-Terry model (Bradley & Terry, 1952) used in Chatbot Arena (Chiang et al., 2024), a popular and dynamic benchmark for AI-powered chatbots. We fit the Rasch model using maximum likelihood estimation as in Chen et al. (2023b) and Maia Polo et al. (2024b).

### 3.3.2 Efficient evaluation with IRT

We can estimate old model scores on new data without reevaluating those models. Let $\mathcal{I}_t$ and $\mathcal{J}_t$ represent sets of non-negative integers corresponding to LMMs and samples at time $t \geq 0$. It is usually the case that $\mathcal{I}_t \subseteq \mathcal{I}_{t+1}$ since the set of available models does not shrink over time, except in cases of deprecation, and $\mathcal{J}_{t_1} \cap \mathcal{J}_{t_2} = \emptyset$ for $t_1 \neq t_2$ because samples are not repeated across different time steps. Let the set of evaluated models at time $t$ be denoted by $\hat{\mathcal{I}}_t$. For $t > 0$, we assume that $\mathcal{I}_t \setminus \mathcal{I}_{t-1}$ is a proper subset of $\hat{\mathcal{I}}_t$, meaning that all newly introduced models are evaluated on the new batch of samples along with some previously existing models. At $t = 0$, we assume that $\hat{\mathcal{I}}_t = \mathcal{I}_t$, meaning all models are evaluated on all samples. Furthermore, we assume that $|\hat{\mathcal{I}}_t|$ is much smaller than $|\mathcal{I}_t|$ when $t > 0$ so computing power and evaluation time can be saved.

Our goal at time $t > 0$ is to estimate the performance of a model $i \notin \hat{\mathcal{I}}_t$ on the set of samples $\mathcal{J}_t$, using only the correctness scores $\mathcal{D}_t = \{Y_{ij} : (i,j) \in \Omega_t\}$, where $\Omega_t \triangleq \cup_{t' \leq t} \hat{\mathcal{I}}_{t'} \times \mathcal{J}_{t'}$. Specifically, we aim to approximate $S_{it} = \frac{1}{|\mathcal{J}_t|} \sum_{j \in \mathcal{J}_t} Y_{ij}$ by estimating its expectation

$$\mathbb{E}[S_{it}] = \frac{1}{|\mathcal{J}_t|} \sum_{j \in \mathcal{J}_t} \mathbb{P}(Y_{ij} = 1; \theta_i, \beta_j). \tag{2}$$

For a moment, let us assume that $\Omega_t$ is known. Using $\mathcal{D}_t$, we can estimate the skill parameters $\theta_i$'s of all models in $\mathcal{I}_t$ and the difficulty parameters $\beta_j$'s of all samples in $\cup_{t' \leq t} \mathcal{J}_{t'}$; we denote these estimates as $\hat{\theta}_i$'s and $\hat{\beta}_j$'s. Finally, we obtain an approximation for equation 2, $\hat{\mathbb{E}}[S_{it}]$, by substituting $\theta_i$ and $\beta_j$'s by their estimates. The estimator $\hat{\mathbb{E}}[S_{it}]$ is known as the Performance-IRT estimator (Maia Polo et al., 2024b;a).

Now, we provide a method to obtain $\Omega_t$ assuming $\Omega_{t-1}$ is given; in summary, we need to decide which models in $\mathcal{I}_{t-1}$ are going to be in $\hat{\mathcal{I}}_t$. Our approach to choosing which models are going to be re-evaluated is inspired by the concept of optimal design of tests (Van der Linden, 2017, Chapter 9) but in which we choose LMMs instead of samples. First, we set a budget $m_t$, representing the maximum number of models to be re-evaluated at time step $t$. Second, assuming that the level of difficulty of the new samples $\mathcal{J}_t$ is not very different from the ones in $\mathcal{J}_{t-1}$, we choose a set of $m_t$ representative samples in $\mathcal{J}_{t-1}$ by ordering $\hat{\beta}_j$'s and choosing equally spaced samples, based on their quantiles, from the 5th to the 95th percentiles; this will give us questions with a variety of difficulties, excluding outliers. For example, if $m_t = 3$ we would choose questions with difficulties in the 5th, 50th, and 95th percentiles. Denote the chosen core set of samples as $\{j_0, \cdots, j_{m_t-1}\}$ and, for each one of these samples $j_k$, we choose a model $i$ in $\mathcal{I}_{t-1}$ such that the following Fisher information criterion

$$F_{j_k}(i) = \mathbb{P}\left(Y_{ij_k} = 1; \hat{\theta}_i, \hat{\beta}_{j_k}\right)\left[1 - \mathbb{P}\left(Y_{ij_k} = 1; \hat{\theta}_i, \hat{\beta}_{j_k}\right)\right]$$

is maximized. The model that maximizes $F_{j_k}$ is maximally informative about the parameter of sample $j_k$ and, consequently, about all samples with similar difficulty levels in the new version of LiveXiv; this will help us estimate the difficulties of new samples. We note that some models in $\mathcal{I}_{t-1}$ might not be available at step $t$, e.g., due to deprecation; when choosing models, we do not consider them, but note that we can still estimate their performance on the new batches of data. Moreover, the model selection procedure can also take convenience into account; for example, if two models have very similar Fisher information, we opt for the one that is cheaper to evaluate.

Our experiments demonstrate that re-evaluating just 5 (or even 3) models per step provides accurate model evaluation. With 50 total models, this approach can reduce computing costs by at least $\times 10$, particularly when selecting less expensive models without significantly impacting $F_{j_k}$. In this paper, we use a maximum of 19 models; thus, re-evaluating 3 to 5 models represents a saving of $\approx 75\%$ to $85\%$ in the number of evaluated models.

## 4 RESULTS & ANALYSIS

This section presents the results obtained on the first version of LiveXiv. First, we start by describing the experimental settings. Then, we present the results and finally conclude with a detailed analysis of our dataset.

### 4.1 EXPERIMENTAL SETTINGS

**Evaluation Protocol:** After the generation of the question-answer pairs from our automated pipeline explained in Section 3, we transform the benchmark to multiple-choice questions. We resort to the 'generate' inference employed extensively by previous works, such as Li et al. (2024d); Huang et al. (2024); Liu et al. (2023d). The model is prompted to choose the letter corresponding to the correct choice and answer with the letter directly. The output letter is then compared with the ground truth and the accuracy is measured. For ease of assimilation and to obtain insights into what type of data the models flourish at, we provide the results from data generated on tables and figures separately. The data generated from figures is labeled as part of Visual Question and Answers (VQA) and the data from the tables is labeled as Table Question and Answers (TQA). Examples for the multiple-choice formulation of the question-answer pairs are added to the Appendix Section A.3.

**Size of dataset:** The first version of our LiveXiv consists of 7328 questions on figures, and 9000 questions on tables, both are generated from 250 papers (25 papers from 10 domains). Overall our first version of the dataset has 16328 questions in total. Thanks to the continual growth in the number of publications in our target domains and the fully automatic nature of our proposed LiveXiv pipeline for benchmark data generation, we will grow LiveXiv by adding an equal-sized large amount of new VQA & TQA data (around 7K VQA and 9K TQA) on a monthly basis.

**Models:** We extensively evaluate our benchmark by employing a total of 17 LMMs. Specifically, we employ 5 models from the LLaVA family of models including LLaVA 1.5-7B and LLaVA 1.5-13B (Liu et al., 2023c), LLaVA-1.6-7B and LLaVA 1.6-34B (Liu et al., 2023b), and LLaVA One-Vision (Li et al., 2024b). Furthermore, we employ IntstructBLIP (Dai et al., 2023), InternVL2-2B and InternVL2-8B (Chen et al., 2023c), InternLM-Xcomposer2-4KHD (Dong et al., 2024b), InternLM-Xcomposer2.5 (Chen et al., 2023c), Mantis (Jiang et al., 2024), Phi3v (Abdin et al., 2024), Idefics2 (Laurençon et al., 2024b) and Idefics3 (Laurençon et al., 2024a), Qwen2-VL (Wang et al., 2024) and API models Claude-Sonnet (cla, 2024) and GPT-4o (OpenAI, 2023) for our evaluations. These models have been chosen because of their varying characteristics and strong performance on multiple current benchmarks. All the models (except GPT-4o and Cloude-Sonnet) are accessed from the huggingface API, which makes our framework modular for an extension to more models as they are being added to the hub in the future.

**Additional LiveXiv versions:** While this section mainly focuses on the analysis of the first version of LiveXiv, new versions are continuously being uploaded to the HuggingFace Hub, at the time of writing, four additional versions exist: one of past ArXiv papers (v0) and three of more recent papers (v2 - v4). Version 2 consists of $18K$ samples, introduces 4 new domains from physics (namely, physics.optics, physics.bio-ph, physics.app-ph, physics.data-an), and includes two additional LMMs (Pixtral and Molmo-7B). Version 3 utilizes multiple generation and filtering models (GPT-4o, Claude-Sonnet, Qwen2-VL-72B) to and increase diversity and measure the effect of potential biases. In Version 4, we restrict diversity to two models, GPT-4o and Claude-Sonnet, but maintain variability by having each model take on different roles each time. In addition, to perform a deeper analysis, we created a variant of version 1 with opposite roles (*i.e.*, Claude as the QA generation model and GPT as the filter model). We found that the question and visual content diversity remained roughly the same which kept the model performance and ranking unaffected. Lastly, version 0 is generated from papers from more than 10 years ago. Our experiments reassure our findings about the performance of our efficient evaluation, strengthening its validity even in the presence of a large time gap between the dataset versions. All the details can be found in Appendix A.4.

### 4.2 EXPERIMENTAL RESULTS

**Results on LiveXiv v1:** We evaluated 17 LMMs across two prominent tasks, VQA and TQA. Table 1 provides a detailed summary of the performance across both tasks. One interesting observation is Claude's superior performance on all tasks. This substantial performance gap suggests that Claude's architecture and underlying methodologies are particularly well-suited for both VQA and TQA tasks. The results align with other relatively close benchmarks, DocVQA (Mathew et al., 2021), ChartQA (Masry et al., 2022) and AI2D (Kembhavi et al., 2016), where we see a similar trend: Claude has significantly higher performance

Table 1: LiveXiv v1 VQA and TQA average accuracy across ArXiv taxonomy. (V is VQA, T-TQA)

| VQA&TQA Acc. | eess.SP | | q-bio.BM | | q-bio.CB | | cs.AI | | eess.SY | | cs.CV | | cs.RO | | q-bio.GN | | cs.LG | | q-bio.TO | | Mean | |
|---|---|---|---|---|---|---|---|---|---|---|---|---|---|---|---|---|---|---|---|---|---|---|
| | V | T | V | T | V | T | V | T | V | T | V | T | V | T | V | T | V | T | V | T | V | T |
| # Samples | 651 | 429 | 900 | 1624 | 840 | 697 | 685 | 1069 | 735 | 472 | 720 | 932 | 672 | 570 | 647 | 1121 | 844 | 1195 | 634 | 894 | 7328 | 9000 |
| InstructBLIP-7B | 21.2 | 18.1 | 25.2 | 16.6 | 19.5 | 20.2 | 24.5 | 21.8 | 23.4 | 18.9 | 21.3 | 20.7 | 22.6 | 22.8 | 24.9 | 16.9 | 24.1 | 18.5 | 21.1 | 18.2 | 23.6 | 19.1 |
| LLaVA-1.5-7B | 29.0 | 24.9 | 27.8 | 20.9 | 29.5 | 25.4 | 31.9 | 23.5 | 30.5 | 25.0 | 31.0 | 21.8 | 34.9 | 24.9 | 29.1 | 22.6 | 29.3 | 24.9 | 32.8 | 25.4 | 30.4 | 23.5 |
| LLaVA-1.6-Mistral-7B | 28.1 | 31.9 | 28.7 | 26.8 | 28.6 | 29.7 | 33.9 | 29.2 | 31.0 | 36.4 | 31.4 | 29.0 | 33.3 | 30.5 | 27.0 | 27.6 | 27.9 | 30.4 | 29.5 | 29.1 | 29.9 | 29.3 |
| Mantis-LLama3-8B | 32.3 | 31.2 | 28.6 | 28.0 | 32.7 | 30.0 | 33.7 | 30.0 | 30.2 | 33.5 | 36.9 | 29.1 | 32.6 | 32.5 | 29.2 | 29.9 | 30.8 | 30.0 | 34.9 | 30.3 | 32.1 | 30.0 |
| LLaVA-1.5-13B | 32.6 | 30.5 | 29.4 | 28.9 | 31.5 | 33.1 | 33.4 | 31.5 | 33.2 | 35.6 | 35.9 | 31.5 | 35.7 | 33.0 | 30.6 | 29.8 | 30.0 | 29.4 | 32.2 | 30.8 | 32.3 | 30.9 |
| Idefics2-8B | 35.6 | 37.1 | 38.4 | 35.9 | 35.9 | 43.2 | 40.7 | 39.2 | 40.5 | 42.8 | 38.6 | 35.0 | 39.6 | 40.0 | 30.3 | 38.7 | 36.9 | 37.0 | 38.8 | 38.9 | 37.6 | 38.2 |
| IXC2-4KHD-7B | 33.0 | 41.1 | 36.7 | 40.8 | 33.0 | 46.8 | 40.1 | 38.4 | 35.8 | 47.2 | 45.7 | 37.0 | 44.5 | 41.9 | 37.9 | 42.8 | 35.8 | 41.3 | 36.1 | 42.6 | 37.7 | 41.5 |
| IXC2.5-7B | 46.2 | 42.3 | 46.1 | 40.5 | 48.2 | 48.4 | 53.3 | 39.8 | 50.5 | 50.8 | 45.1 | 39.2 | 47.0 | 44.6 | 47.9 | 47.5 | 49.4 | 36.7 | 46.8 | 40.7 | 48.1 | 42.1 |
| InternVL2-2B | 48.4 | 42.7 | 48.1 | 44.2 | 50.4 | 53.7 | 53.4 | 48.9 | 50.5 | 58.3 | 46.3 | 44.7 | 54.2 | 51.6 | 48.4 | 52.0 | 48.2 | 49.2 | 50.9 | 52.0 | 49.8 | 49.1 |
| LLaVA-1.6-34B | 48.4 | 49.5 | 45.6 | 49.4 | 47.4 | 55.7 | 55.9 | 49.2 | 52.5 | 59.7 | 51.8 | 48.9 | 54.9 | 50.7 | 47.9 | 49.2 | 47.9 | 46.6 | 50.2 | 50.8 | 50.0 | 50.2 |
| Idefics3 | 54.4 | 47.2 | 50.6 | 48.1 | 52.3 | 55.5 | 57.2 | 48.9 | 57.0 | 57.8 | 53.3 | 47.2 | 54.6 | 51.9 | 51.5 | 51.7 | 51.5 | 48.0 | 56.6 | 51.8 | 53.7 | 50.2 |
| LLaVA-OneVision-7B | 53.1 | 46.2 | 49.7 | 47.8 | 51.8 | 53.9 | 57.2 | 50.3 | 52.8 | 57.8 | 57.2 | 47.7 | 57.6 | 53.2 | 51.6 | 51.2 | 51.1 | 50.5 | 59.1 | 52.7 | 53.9 | 50.6 |
| Phi3v | 60.1 | 51.4 | 54.4 | 48.7 | 59.9 | 54.8 | 64.5 | 52.8 | 61.8 | 57.8 | 56.0 | 48.7 | 58.5 | 51.4 | 58.9 | 51.0 | 56.0 | 51.5 | 58.2 | 56.2 | 58.7 | 51.8 |
| GPT-4o | 64.1 | 50.7 | 55.9 | 51.8 | 58.8 | 56.2 | 62.9 | 54.3 | 64.4 | 62.3 | 60.1 | 50.8 | 60.3 | 56.1 | 55.2 | 56.3 | 59.0 | 55.1 | 64.4 | 55.0 | 60.3 | 54.5 |
| InternVL2-8B | 64.5 | 57.5 | 56.9 | 57.5 | 61.4 | 65.3 | 67.0 | 57.5 | 65.3 | 67.2 | 59.9 | 60.1 | 65.3 | 61.8 | 58.4 | 60.8 | 61.4 | 59.1 | 65.6 | 61.4 | 62.3 | 60.2 |
| Qwen2-VL | 68.0 | 60.3 | 62.4 | 59.6 | 71.8 | 67.3 | 67.2 | 59.7 | 69.3 | 70.1 | 63.3 | 62.6 | 64.6 | 64.6 | 64.5 | 61.1 | 63.7 | 59.2 | 71.9 | 65.0 | 66.6 | 62.1 |
| Claude-Sonnet | 78.9 | 84.0 | 72.3 | 81.2 | 77.4 | 80.3 | 77.7 | 84.5 | 78.4 | 85.6 | 69.9 | 84.0 | 74.1 | 86.5 | 72.9 | 82.9 | 76.4 | 86.4 | 75.9 | 82.3 | 75.4 | 83.5 |

Table 2: Performance change between LiveXiv (v1) and a manually verified subset averaged across all evaluated models. LiveXiv is robust, thanks to excessive filtering steps which keep the labeling errors low.

| | LiveXiv | Verified Subset | Absolute Avg. |
|---|---|---|---|
| **VQA** | 46.734 | 47.273 | 2.336 |
| **TQA** | 45.101 | 46.028 | 2.105 |

over the runner-up models such as Qwen2-VL, GPT-4o and InternVL2-8B. See Table 4 for more details. However, a notable caveat is that Claude plays an integral role in the question-filtering process, which may introduce a potential bias in favor of questions it is predisposed to solve effectively. This implies that while Claude's overall performance remains strong, the evaluation might not fully reflect its robustness to novel or more diverse question types outside the scope of this filtering. Surprisingly, GPT-4o has low performance. To understand the root cause of this observation, we experimented with various hyper-parameters such as temperature, image resolution, and diverse textual prompts. Nevertheless, these all yielded similar results.

We further observe that newer models, such as InternVL2-8B and Qwen2-VL, consistently outperform older models like LLaVA-1.6 and Idefics2, suggesting rapid advancements in LMM development over the past few months. This trend highlights the continual improvement in both architecture and training paradigms, leading to better generalization across multi-modal tasks.

Zooming into the domain-specific performance using an ArXiv-based taxonomy, we evaluate each model's effectiveness in distinct scientific fields such as biology, electrical engineering, and mathematics. Our results show that certain models, particularly the newer architectures, exhibit a higher degree of robustness across diverse domains, highlighting that the models' training data might already have potential contamination issues. Conversely, for VQA, models in the Intern-VL2 and the LLaVA families appear to be more sensitive to domain shifts, performing inconsistently across different scientific areas, as oppose to the more recent models like Qwen2-VL, Claude and GPT-4o, see Figures 6, 7 for more details. For TQA, it's not the case, probably since the questions test more specific skills such a retrieval and arithmetic manipulations, see Figures 8, 9. This domain-specific sensitivity emphasizes the need for further refinements in LMMs, especially when applied to specialized scientific knowledge domains. Overall, this analysis not only underscores the ongoing evolution of LMMs but also highlights areas for further investigation, especially concerning model adaptability to diverse content domains and the potential biases introduced by models.

**Contamination free effect:** Interestingly, focusing on new data that came after the LMMs were trained, allows LiveXiv to provide a new, contamination-free, perspective on the relative performance ranking between strong LMMs. For example, taking the official results from original publications and computing the average ranking of the evaluated LMMs over the established DocVQA (Mathew et al., 2021), ChartQA (Masry et al., 2022) and AI2D (Kembhavi et al., 2016) benchmarks, and comparing those to the average rankings provided by LiveXiv, we observe some significant ranking changes. *e.g.*, IXC2.5 and IXC2-4KHD drop over 4 points in average ranking. See Table 5 in the appendix for details.

**Performance on manually filtered dataset:** To further verify our proposed automated question-answer generation and filtering methodology and to obtain a measure of errors in the generated data, we manually verified a subset of 1000 samples (500 for both, VQA and TQA) and evaluated all models on this subset. Table 2 presents the results for VQA and TQA on the filtered subset. We see that on average

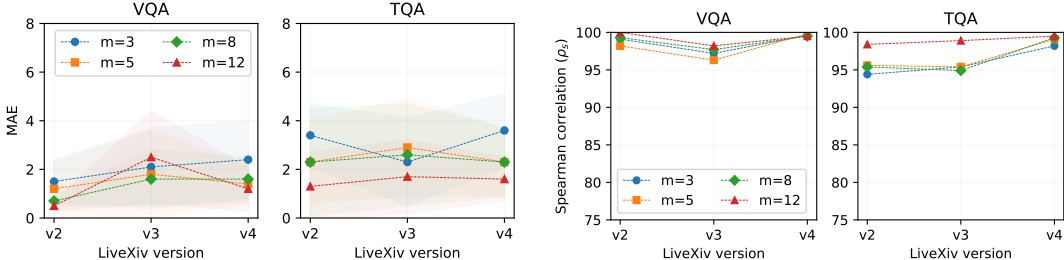

Figure 4: Average prediction errors and rank correlations for overall performance. We report MAE ($\pm$ mean absolute deviation) for non-re-evaluated models and Spearman's rank correlation across 19 LMMs on different LiveXiv versions. Re-evaluating just 3–5 models is generally sufficient for accurate performance prediction.

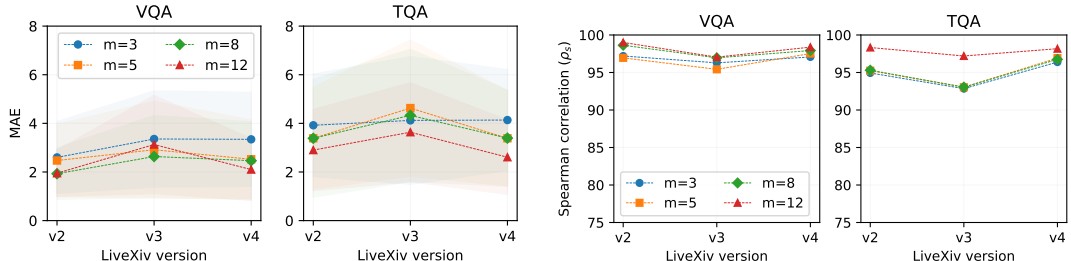

Figure 5: Average prediction errors and rank correlations when we predict performance for each domain separately. We report average MAE ($\pm$ average mean absolute deviation) for non-re-evaluated models and average Spearman's rank correlation across 19 LMMs on different LiveXiv versions. A worse performance LiveXiv v3 can be explained by a smaller number of evaluations.

the performance only fluctuates by $2.3\%$ and $2.1\%$ for VQA and TQA when comparing the results obtained by all the models on the entire dataset and the manually verified subset. These results hint that our automated question-answer generation pipeline and the filtering methodology is quite robust. Detailed results can be found at the Appendix, Tables 6 and 7.

**Efficient evaluations of LMMs:** In this section, we empirically validate the effectiveness of our proposed efficient re-evaluation method for LMMs using LiveXiv v1-v4. Dynamic benchmarks like LiveXiv present a challenge in terms of evaluation costs since each time a new version of the benchmark is released, all models should be re-evaluated on the updated data. This process, however, can become computationally prohibitive when dealing with numerous models. Our goal is to demonstrate that by re-evaluating only a small subset of models on a new version of LiveXiv, we can still reliably predict the performance of the remaining models. For this experiment, we evaluate 17 models on the first version of LiveXiv but only re-evaluate $m \in \{3,5,8,12\}$ models on the subsequent versions of the benchmark using the model selection methodology detailed in Section 3.3; we focus on VQA or TQA (but not both simultaneously) and consider all ArXiv domains. From LiveXiv v2 onwards, two new extra models were added to the analysis; these do not count towards the budget of $m$ models at v2.

At every step, an IRT model is fitted to the full observed data and we predict the performance of the non-re-evaluated models on the full VQA and TQA versions of LiveXiv and on each ArXiv domain using empirical versions of equation 2. Figure 4 presents the results for the case we aim to predict the performance of the non-re-evaluated models on the full VQA and TQA versions of LiveXiv while Figure 5 presents the averages of errors and correlations across domains. They report both the mean absolute error (MAE) ($\pm$ mean absolute deviation) for the non-re-evaluated models when predicting their accuracy and Spearman's rank correlation across all 19 LMMs on different LiveXiv versions when comparing real accuracy and predicted accuracy. These results suggest that re-evaluating just 3 or 5 models is likely to be sufficient for accurately predicting the performance of the remaining models and the ranking of all models, especially when our focus is predicting the overall performances. This last observation makes sense since the full benchmarks contain many more data points when compared to individual domains and, in that case, individual prediction errors tend to cancel out in the empirical version of equation 2. For the same reason, we ended up having a poorer performance on LiveXiv v3, which has fewer data points; as detailed in Appendix A.4.4, LiveXiv v3 has questions generated by three models as part of an ablation study, however, we ultimately use the questions generated by GPT-4o and Claude-Sonnet in this study to keep procedures more standardized with v4.

In Appendix A.5, we present additional results to further validate the effectiveness of our method. Specifically, we (i) show a detailed error analysis of the results for LiveXiv v4 (from Figures 4/5), (ii)

Table 3: LiveXiv accuracy (%) on different categories of question and partitions averaged over all evaluated models.

| | Data Analysis | Reasoning | Attribute | Localization | Reading | Arithmetic | Charts | Block Diagram | Qualitative |
|---|---|---|---|---|---|---|---|---|---|
| VQA | 46.93 | 47.95 | 46.18 | 41.91 | 47.83 | 46.87 | 44.17 | 52.69 | 48.60 |
| TQA | 46.02 | 63.61 | 68.69 | 51.66 | 59.35 | 35.56 | - | - | - |

test a situation with high distribution shift using a hypothetical version of LiveXiv consisting of ArXiv papers from 2010 (v0), (iii) validate our efficient evaluation method when questions are also generated by Qwen2-VL-72B in v3, and (iv) test our approach on MM-LiveBench (Zhang et al., 2024b).

## 4.3 ANALYSIS AND ABLATIONS

To analyze various aspects of LiveXiv we provide an extensive ablation study. We start by providing an analysis of the results from different models obtained w.r.t the language content partitions, then provide results for different models w.r.t the visual data partitions.

**Language analysis - performance according to question type.** To discover error slices of models for an analysis of mistakes they commonly make, we classify the questions present in the benchmark into one of the following categories: reasoning, data analysis, reading, localization, attribute, and arithmetic. To achieve this classification, we employ the Llama-3.1 (Meta, 2024) LLM and prompt the model with the question and the list of categories to choose for this question. The prompt is provided in the Appendix Figure 21. Table 3 summarizes the results for all the models. We see that the performance of these models on the arithmetic partition is the lowest on average as compared to other partitions highlighting room for potential improvement. We also provide the detailed results for all models on these partitions for VQA and TQA in Tables 9 and 10 of the Appendix.

**Vision analysis - performance according to figure type.** For a more fine-grained analysis of LMM performance on different types of visual data present in our benchmark, we first categorize the data through Meta-Prompting for CLIP, proposed by Mirza et al. (2024), in a zero-shot classification setup. Specifically, we classify the image content into three categories of figures: Block diagrams, Qualitative visual results, and Charts. We summarize the results in Table 3. Detailed results for each model's performance can be found in Table 8 in the Appendix. The results reveal a significant variance in performance across figure types for nearly all models. In most cases, block diagrams are the most favorable category for models. However, InternLM-Xcomposer2-4KHD-7B (Dong et al., 2024b) stands out by achieving the highest accuracy on Qualitative figures. Overall, Charts emerge as the most challenging figure type on average, suggesting a lack of sufficient examples in the training data for this category. This kind of analysis can be further expanded to include more categories and discover error slices on which different models struggle so that potential targeted improvements can be designed for these models to mitigate the shortcomings.

**Diversity & Difficulty** The visual content of scientific papers (figures and tables) evolves over time, and we aim to ensure our questions remain diverse across dataset versions. To achieve this, we use an LLM (Llama-3.1-70B Meta (2024)) to classify questions into predefined categories, allowing us to monitor and maintain diversity. Table 18 confirms that LiveXiv maintains question diversity across versions. Additionally, we assess question difficulty by analyzing how many models answer each question correctly. Figure 10 shows that the difficulty distribution for both VQA and TQA tasks remains stable across dataset versions.

## 5 LIMITATIONS AND CONCLUSIONS

**Limitations.** LiveXiv relies on capable proprietary LMMs in order to be fully automatic, and with high quality. However, relying on proprietary LMMs is a limitation since we do not have full control over the models, they can change through time and might affect LiveXiv. Nevertheless, we commonly expect them to continuously improve leading to a positive impact on LiveXiv effectiveness.

**Conclusions.** We propose LiveXiv, an ever-evolving, fully automatic, multi-modal benchmark focused on scientific domains to tackle test set contamination issues and consequently allow a new (contamination-free) perspective on relative ranking of advanced LMMs. We utilize ArXiv, as the data source, carefully and extensively crafting a quality dataset to evaluate LMMs. To significantly reduce the computational and logistical overhead of maintaining the dataset throughout time and models, we propose an efficient evaluation method that can save more than 70% of the evaluated models on each dataset version. Our method can be extended to other archives such as BioRXiv to extend our dataset to new domains. For future work, we propose enhancing LMM evaluation by incorporating free-form questions to assess generative abilities and complex question types to evaluate reasoning capabilities.

## 6 ETHICS STATEMENT

This work introduces LiveXiv, a live multi-modal benchmark for evaluating LMMs using scientific ArXiv papers. By relying solely on publicly available ArXiv manuscripts with proper licenses, we ensure compliance with copyright and distribution policies. The automated generation of Visual Question Answering (VQA) and Table Question Answering (TQA) pairs enables scalable evaluation of LMMs without human involvement, minimizing the risk of human biases in data collection. However, we acknowledge the potential for unintentional biases within the models or dataset itself. Continuous evaluation and refinement are necessary to mitigate these biases and promote the responsible deployment of LMMs in wider applications.

## 7 ACKNOWLEDGMENTS

This research was partially supported by the TAD center at Tel Aviv University. The authors thank Eli Schwartz for his valuable feedback and assistance.

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

# A  APPENDIX

## A.1  RELATED WORKS

### A.1.1  EFFICIENT BENCHMARKS

With the increasing amount of tasks and samples in current benchmarks, evaluation of the full suite is time-consuming and cost-intensive. Efforts are underway to develop efficient benchmarks that reduce computation costs without sacrificing reliability. For LLMs, Perlitz et al. (2023) proposed the first systematic study of the effects of language model benchmark designs on reliability and efficiency, and applied efficient benchmark practices on the HELM benchmark (Liang et al., 2022), leading to $\times 100$ computation reduction with minimal loss on reliability. Lifelong benchmarks (Prabhu et al., 2024) has an ever-expanding pool of test samples for the categories in CIFAR10 (Krizhevsky & Hinton, 2009) and ImageNet (Deng et al., 2009); to make this design economically feasible, it reuses past model evaluations on a sample set through dynamic programming to enable efficient evaluation of new incoming models, drastically reducing the evaluation cost. Most related to our work, tinyBenchmarks (Maia Polo et al., 2024a) and PromptEval (Maia Polo et al., 2024b) propose using Item Response Theory (IRT) (Lord et al., 1968) to estimate the performance of LLMs on unseen samples, making efficient evaluation possible by only conducting a small fraction of the total number of evaluations. Inspired by the last two works, we leverage IRT to estimate the performance of older models in new batches of data. More specifically, at each version of LiveXiv, we choose a small core set of models ($\leq 5$) previously added to the leaderboard and re-evaluate them on the new data. Depending on their responses to the new samples, we estimate the performance of the remaining old models on the new benchmark version.

### A.1.2  LLM CONTAMINATION-FREE BENCHMARKS

For LLMs, LMSys Chatbot Arena (Chiang et al., 2024) and AI2 WildVision (Lu et al., 2024) create a user-focused platform that provides contamination-free environment for proper evaluations. However, it is expensive to collect tens of thousands of human preferences on the compared language models. Furthermore, Seal Benchmark (AI, 2024) proposes private questions paired with human evaluations. Srivastava et al. (2024) update the questions in the MATH dataset (Hendrycks et al., 2021) by changing numbers in the math questions. LiveBench White et al. (2024) collects frequently updated questions from diverse information sources *e.g.* math competitions, arXiv papers and news articles and more challenging versions of existing benchmark tasks. Concurrently, LiveCodeBench (Jain et al., 2024) contributes a live benchmark on broader code-related capabilities. Note that these datasets focus on language data only.

## A.2  ANALYSIS & ABLATIONS

Table 4: **Average results for relatively close benchmarks (DocVQA, ChartQA and AI2D).** We can see that Claude, GPT4o, Qwen2-VL and InternVL2 are the top models. The overall ordering is align with our benchmark.

|                      | ChartQA | DocVQA | AI2D |
|----------------------|---------|--------|------|
| InstructBLIP-7B      | 10.9    | 74     | 40.6 |
| LLaVA-1.6-Mistral-7B | 51.8    | 72.2   | 69   |
| Mantis               | 42.9    | -      | 60.4 |
| LLaVA-1.5-7B         | 17.8    | 74.4   | 55.5 |
| LLaVA-1.5-13B        | 18.2    | 77.5   | 70   |
| Idefics2             | -       | 74     | 72.3 |
| InternVL2-2B         | 76.2    | 86.9   | 74.1 |
| IXC2-4KHD-7B         | 81      | 90     | 81   |
| IXC2.5-7B            | 82.2    | 90.9   | 81.6 |
| LLaVA-OneVision-7B   | 80      | 83.7   | 81.4 |
| Phi3v                | 72      | 84.9   | 77.8 |
| Idefics3             | -       | 87.7   | 76.5 |
| LLaVA-1.6-34B        | 67.6    | 84     | 78.9 |
| GPT-4o               | 85.7    | 92.8   | 94.2 |
| Qwen2-VL             | 83      | 94.5   | 83   |
| InternVL2-8B         | 83.3    | 91.6   | 83.8 |
| Claude-Sonnet        | **90.8**| **95.2**| **94.7** |

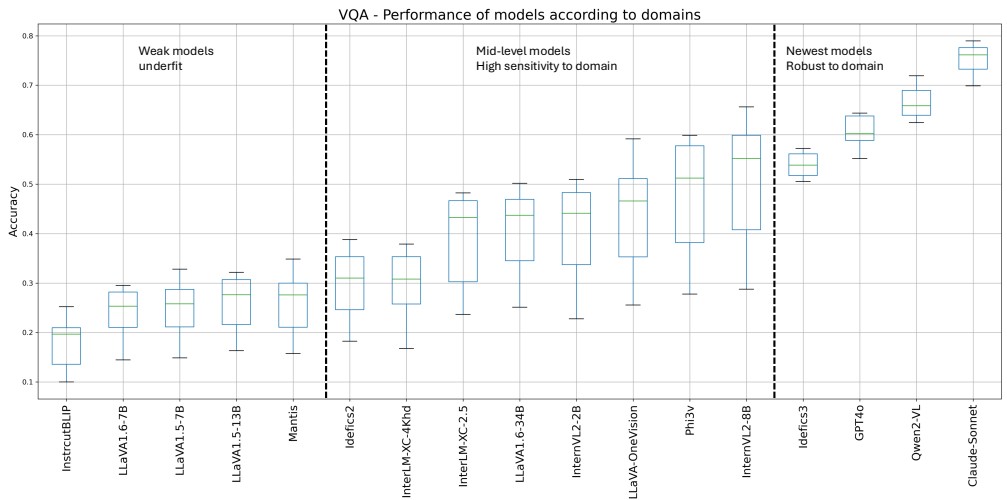

Figure 6: **Domain sensitivity according to domains.** We visualize the performance of each model across all domains. Clear trends revealed where old models or models with a small LLM are "under-fitting" and perform worse across all domains. In the middle we have the mid-level models that are sensitive to the domain, indicating their lack of generalization across domain without any additional training. Lastly the newest models (open-source and proprietary) are robust to domain shifts and present a stable performance across the domains.

Table 5: Average ranking on static benchmarks (ChartQA, DocVQA and AI2D) and LiveXiv (v1). We can see from the ranking difference column that some models have a significant drop (negative difference) in the relative ranking in LiveXiv compared to the static datasets. The gap is highlighting a potential risk of test data contamination when using static (frozen in time) benchmark datasets.

| Model | Static datasets | LiveXiv | Difference (static - livexiv) |
|---|---|---|---|
| InstructBLIP-7B | 15.33 | 17.00 | -1.67 |
| LLaVA-1.6-Mistral-7B | 13.67 | 16.00 | -2.33 |
| Mantis | 14.50 | 14.50 | 0.00 |
| LLaVA-1.5-7B | 14.33 | 14.50 | -0.17 |
| LLaVA-1.5-13B | 12.67 | 13.00 | -0.33 |
| Idefics2 | 13.00 | 12.00 | 1.00 |
| InternVL2-2B | 9.00 | 10.00 | -1.00 |
| IXC2-4KHD-7B | 6.33 | 10.50 | **-4.17** |
| IXC2.5-7B | 5.00 | 9.50 | **-4.50** |
| LLaVA-OneVision-7B | 8.00 | 6.50 | 1.50 |
| Phi3v | 9.00 | 6.00 | 3.00 |
| Idefics3 | 8.50 | 6.50 | 2.00 |
| LLaVA-1.6-34B | 9.33 | 6.50 | 2.83 |
| GPT-4o | 2.33 | 4.00 | **-1.67** |
| Qwen2-VL | 3.33 | 3.00 | 0.33 |
| InternVL2-8B | 3.33 | 2.00 | 1.33 |
| Claude-Sonnet | 1.00 | 1.00 | 0.00 |

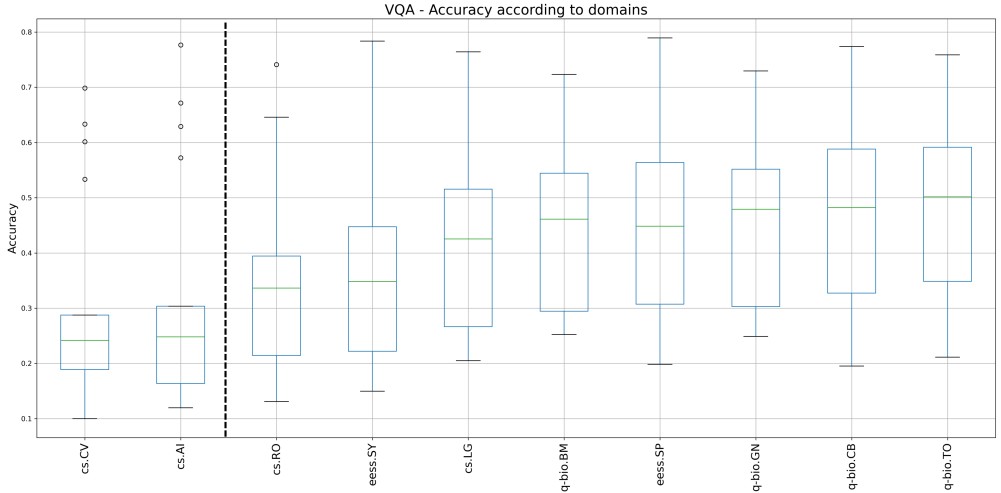

Figure 7: **LMMs performance based on domain.** To complement our analysis form Figure 6 we visualize the statistical properties of each domain. One clear trend is that across all modesl, the performance on cs.CV and cs.AI is the most concentrated, hinting lower variance between models.

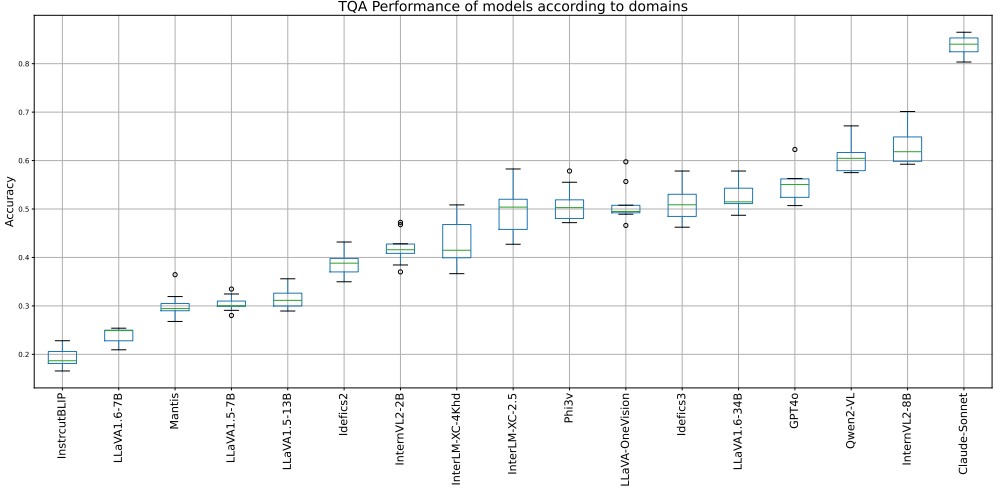

Figure 8: **Domain sensitivity according to domains.** As opposed to the high variance some models demonstrated in Figure 6, in TQA the tasks and he visual content are more limited thus shrunken the performance variance greatly.

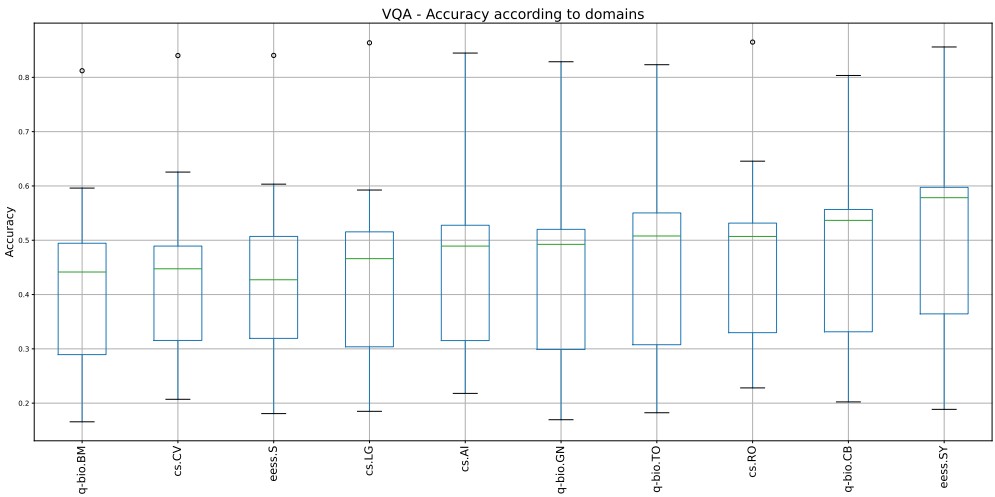

Figure 9: **LMMs performance based on domain.** The domains are very similar in their statistical properties showing high variance in performance. This is probably due to wide range of models that differ significantly in their performance.

### A.2.1 PERFORMANCE CHANGE COMPARED TO MANUALLY CURATED SUBSET

Table 6: VQA Performance change between LiveXiv (v1) and a manually curated subset (500 examples).

| Model | LiveXiv (%) | Manual (%) | Performance Change | LiveXiv Rank | Manual Rank | Change |
|---|---|---|---|---|---|---|
| InstructBLIP-7B | 23.216 | 21.346 | -1.87 | 17 | 17 | 0 |
| InternLM-Xcomposer2.5-7B | 47.839 | 50.769 | 2.93 | 10 | 9 | 1 |
| InternVL2-8B | 61.558 | 66.154 | 4.596 | 3 | 3 | 0 |
| LLaVA-1.6-Mistral-7B | 29.163 | 26.346 | -2.816 | 16 | 16 | 0 |
| LLaVA-OneVision-Qwen2-7B | 52.864 | 56.154 | 3.29 | 7 | 6 | 1 |
| LLaVA-1.5-13B | 31.859 | 30.385 | -1.475 | 14 | 13 | 1 |
| LLaVA-1.5-7B | 29.983 | 28.654 | -1.329 | 15 | 14 | 1 |
| LLaVA-1.6-34B | 49.196 | 53.269 | 4.073 | 9 | 7 | 2 |
| Mantis-LLama3-8B | 32.094 | 28.654 | -3.44 | 13 | 14 | -1 |
| Phi3v | 58.141 | 58.654 | 0.513 | 5 | 5 | 0 |
| InternLM-Xcomposer2-4KHD-7B | 36.801 | 33.654 | -3.147 | 12 | 12 | 0 |
| Idefics2-8B | 36.851 | 36.731 | -0.12 | 11 | 11 | 0 |
| InternVL2-2B | 49.548 | 48.654 | -0.894 | 8 | 10 | -2 |
| Claude-Sonnet | 75.942 | 79.615 | 3.673 | 1 | 1 | 0 |
| Qwen2-VL | 66.248 | 71.346 | 5.098 | 2 | 2 | 0 |
| GPT-4o | 60.303 | 60.577 | 0.274 | 4 | 4 | 0 |
| Idefics3 | 52.881 | 52.692 | -0.189 | 6 | 8 | -2 |
| Average (absolute) change | | | 2.336 | | | |

Table 7: TQA Performance change between LiveXiv (v1) and a manually curated subset (500 examples).

| Model | LiveXiv (%) | Manual (%) | Performance Change | LiveXiv Rank | Manual Rank | Change |
|---|---|---|---|---|---|---|
| InstructBLIP-7B | 19.1 | 18.5 | -0.6 | 17 | 17 | 0 |
| InternLM-Xcomposer2.5-7B | 49.1 | 45.9 | -3.2 | 9 | 9 | 0 |
| InternVL2-8B | 62.1 | 65.3 | 3.2 | 2 | 2 | 0 |
| LLaVA-1.6-Mistral-7B | 23.5 | 23.2 | -0.3 | 16 | 16 | 0 |
| LLaVA-OneVision-Qwen2-7B | 50.2 | 51.6 | 1.4 | 7 | 8 | -1 |
| LLaVA-1.5-13B | 30.9 | 31.2 | 0.3 | 13 | 13 | 0 |
| LLaVA-1.5-7B | 30.0 | 29.6 | -0.3 | 14 | 14 | 0 |
| LLaVA-1.6-34B | 51.8 | 52.2 | 0.4 | 5 | 7 | -2 |
| Mantis-LLama3-8B | 29.3 | 28.0 | -1.3 | 15 | 15 | 0 |
| Phi3v | 50.2 | 54.1 | 4.0 | 7 | 6 | 1 |
| InternLM-Xcomposer2-4KHD-7B | 42.1 | 41.7 | -0.4 | 10 | 11 | -1 |
| Idefics2-8B | 38.2 | 42.0 | 3.8 | 12 | 10 | -2 |
| InternVL2-2B | 41.5 | 39.5 | -2.0 | 11 | 12 | -1 |
| Claude-Sonnet | 83.5 | 89.2 | 5.6 | 1 | 1 | 0 |
| Qwen2-VL | 60.2 | 58.3 | -1.9 | 3 | 3 | 0 |
| GPT-4o | 54.5 | 55.7 | 1.3 | 4 | 5 | -1 |
| Idefics3 | 50.6 | 56.4 | 5.7 | 6 | 4 | 2 |
| Average (absolute) change | | | 2.105 | | | |

### A.2.2 FIGURE TYPE

We provide all the details for VQA performance according to figure type content in Table 8. We divide the performance to the following figure types: "Chart", "Block Diagram" and "Qualitative".

### A.2.3 QUESTION CATEGORY

In Tables 9 and 10 we provide detailed results for VQA and TQA performance according to the category of the questions as classified by an LLM. We divide the performance to the following categories: "Data Analysis", "Attribute", "Reasoning", "Reading", "Localization" and "Arithmetic"

### A.2.4 QUESTIONS' DIFFICULTY

We provide additional analyses regarding the diverse difficulties of LiveXiv's generated questions. Figure 10 demonstrates that for all LiveXiv tasks and for both the first and second versions, a wide range of difficulties is present, where most of the questions concentrate in the middle, suggesting our dataset is indeed challenging.

Table 8: Performance of LMMs over different figure types from the LiveXiv v1 VQA set (the amount of samples for each figure type is in brackets).

| Model | Chart (4354) | block_diagram (2110) | Qualitative (864) |
|---|---|---|---|
| InstructBLIP-7B | 22.9 | 22.8 | 22.5 |
| InternLM-Xcomposer2.5-7B | 46.7 | 53.5 | 41.7 |
| InternVL2-8B | 59.1 | 68.4 | 63.8 |
| LLaVA-1.6-Mistral-7B | 27.3 | 33.6 | 33.4 |
| LLaVA-OneVision-Qwen2-7B | 48.7 | 62.8 | 57.9 |
| LLaVA-1.5-13B | 29.3 | 35.4 | 40.2 |
| LLaVA-1.5-7B | 28.1 | 33.1 | 36.0 |
| LLaVA-1.6-34B | 44.6 | 60.0 | 52.7 |
| Mantis-LLama3-8B | 29.3 | 36.2 | 36.1 |
| Phi3v | 56.1 | 65.5 | 55.2 |
| InternLM-Xcomposer2-4KHD-7B | 32.9 | 43.7 | 47.1 |
| Idefics2-8B | 34.3 | 43.0 | 41.1 |
| InternVL2-2B | 47.2 | 54.9 | 50.0 |
| Claude-Sonnet | 73.7 | 81.3 | 69.1 |
| Qwen2-VL | 63.1 | 73.9 | 66.0 |
| GPT-4o | 56.5 | 68.6 | 59.5 |
| idefics3 | 51.1 | 59.0 | 54.1 |

Table 9: LiveXiv v1 VQA Performance by Question Categories (the amount of samples for each category is in brackets).

| Model | Data Analysis (2291) | Reasoning (872) | Attribute (903) | Localization (1596) | Reading (1470) | Arithmetic (154) |
|---|---|---|---|---|---|---|
| InstrcutBLIP | 21.16 | 29.29 | 22.77 | 31.25 | 23.87 | 23.01 |
| InterLM-XC-2.5 | 47.52 | 43.43 | 48.51 | 31.25 | 50.46 | 47.28 |
| InternVL2-8B | 61.74 | 63.64 | 65.35 | 56.25 | 62.54 | 62.41 |
| LLaVA1.6-7B | 28.91 | 31.31 | 30.69 | 12.50 | 30.00 | 30.43 |
| LLaVA-OneVision | 52.57 | 54.55 | 60.40 | 56.25 | 56.76 | 52.85 |
| LLaVA1.5-13B | 32.92 | 25.25 | 25.74 | 43.75 | 31.85 | 32.58 |
| LLaVA1.5-7B | 30.56 | 28.28 | 29.70 | 25.00 | 29.71 | 30.89 |
| LLaVA1.6-34B | 50.87 | 43.43 | 45.54 | 37.50 | 49.60 | 50.00 |
| Mantis | 32.50 | 33.33 | 30.69 | 43.75 | 31.97 | 31.80 |
| Phi3v | 58.05 | 63.64 | 59.41 | 43.75 | 58.15 | 59.07 |
| InterLM-XC-4Khd | 37.51 | 43.43 | 39.60 | 18.75 | 38.67 | 37.09 |
| Idefics2 | 37.79 | 39.39 | 35.64 | 31.25 | 39.54 | 36.34 |
| InternVL2-2B | 50.26 | 46.46 | 44.55 | 43.75 | 51.45 | 48.62 |
| Claude-Sonnet | 74.78 | 78.79 | 67.33 | 81.25 | 75.49 | 75.64 |
| Qwen2-VL | 66.93 | 66.67 | 68.32 | 43.75 | 68.73 | 65.01 |
| GPT4o | 61.41 | 60.61 | 59.41 | 62.50 | 59.83 | 59.76 |
| Idefics3 | 52.34 | 63.64 | 51.49 | 50.00 | 54.51 | 54.00 |

Table 10: LiveXiv v1 TQA Performance by Question Categories (the amount of samples for each category is in brackets).

| Model | Data Analysis (2582) | Reasoning (123) | Attribute (121) | Localization (23) | Reading (2127) | Arithmetic (3934) |
|---|---|---|---|---|---|---|
| InstructBLIP-7B | 24.7 | 27.6 | 34.7 | 43.5 | 20.4 | 13.6 |
| InternLM-Xcomposer2.5-7B | 54.9 | 75.6 | 79.3 | 60.9 | 74.6 | 29.7 |
| InternVL2-8B | 58.6 | 73.2 | 78.5 | 65.2 | 79.8 | 54.1 |
| LLaVA-1.6-Mistral-7B | 30.4 | 39.0 | 52.1 | 30.4 | 32.0 | 13.0 |
| LLaVA-OneVision-Qwen2-7B | 47.5 | 72.4 | 80.2 | 60.9 | 69.1 | 40.3 |
| LLaVA-1.5-13B | 31.6 | 49.6 | 47.9 | 30.4 | 32.4 | 28.5 |
| LLaVA-1.5-7B | 30.8 | 39.0 | 42.1 | 43.5 | 31.6 | 27.9 |
| LLaVA-1.6-34B | 46.4 | 69.1 | 76.9 | 47.8 | 66.8 | 46.1 |
| Mantis-LLama3-8B | 28.7 | 39.0 | 46.3 | 21.7 | 32.4 | 27.3 |
| Phi3v | 52.2 | 76.4 | 77.7 | 60.9 | 72.4 | 35.2 |
| InternLM-Xcomposer2-4KHD-7B | 45.4 | 69.1 | 77.7 | 60.9 | 66.2 | 25.0 |
| Idefics2-8B | 35.0 | 57.7 | 57.9 | 43.5 | 51.1 | 32.4 |
| InternVL2-2B | 36.9 | 57.7 | 70.2 | 30.4 | 62.2 | 32.1 |
| Claude-Sonnet | 85.1 | 90.2 | 91.7 | 87.0 | 91.0 | 78.1 |
| Qwen2-VL | 64.6 | 86.2 | 90.1 | 65.2 | 82.0 | 44.0 |
| GPT-4o | 57.5 | 79.7 | 86.8 | 69.6 | 73.0 | 40.6 |
| Idefics3 | 52.0 | 79.7 | 77.7 | 56.5 | 72.0 | 36.6 |

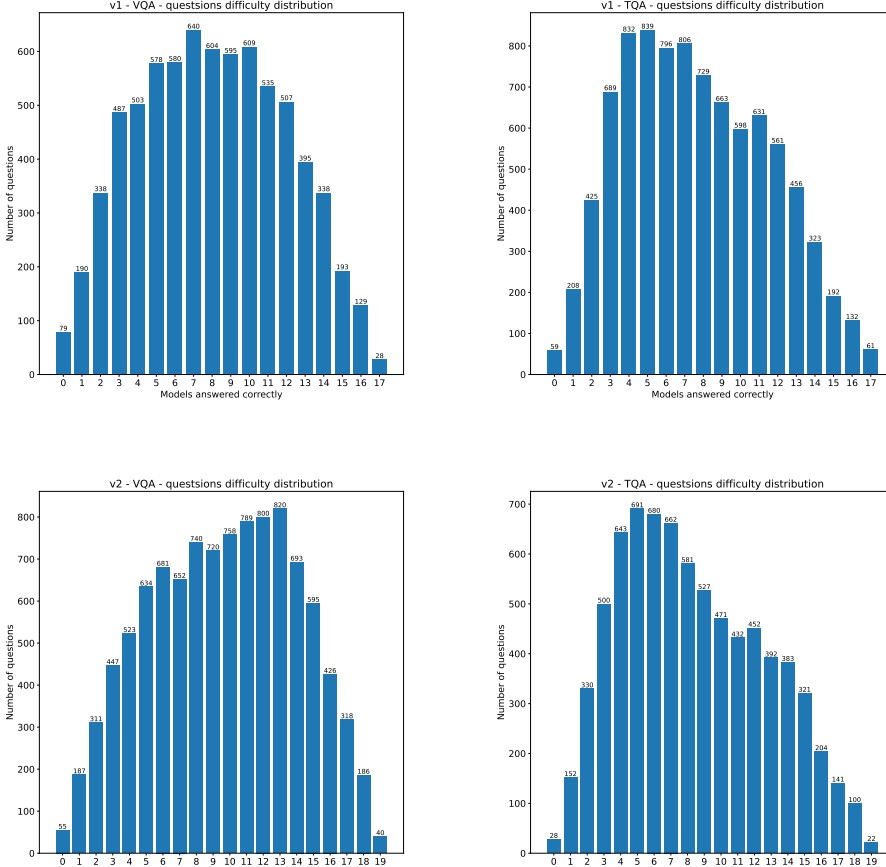

Figure 10: **LiveXiv questions' difficulty diversity.** LiveXiv is composed of questions at varying levels of difficulty. Each row shows a version of the dataset on both of its task (VQA and TQA). LiveXiv demonstrates questions in a wide range of questions' difficulties, from questions that no model was able to answer correctly up to questions that all models were able to easily answer, The center mass consternates in the middle, hinting the dataset is indeed challenging and the fact that the difficulty distribution remains the same for both the first version and second version suggests the stability of our diverse and challenging dataset.

## A.3 DETAILED EXAMPLES FOR VQA AND TQA GENERATION

Here we present full and detailed examples of our flow from ArXiv papers until constructing verified multi-choice Q&A. Figure 11 shows the full example for generating questions from figures (VQA). Figure 12 shows the full examples for TQA.

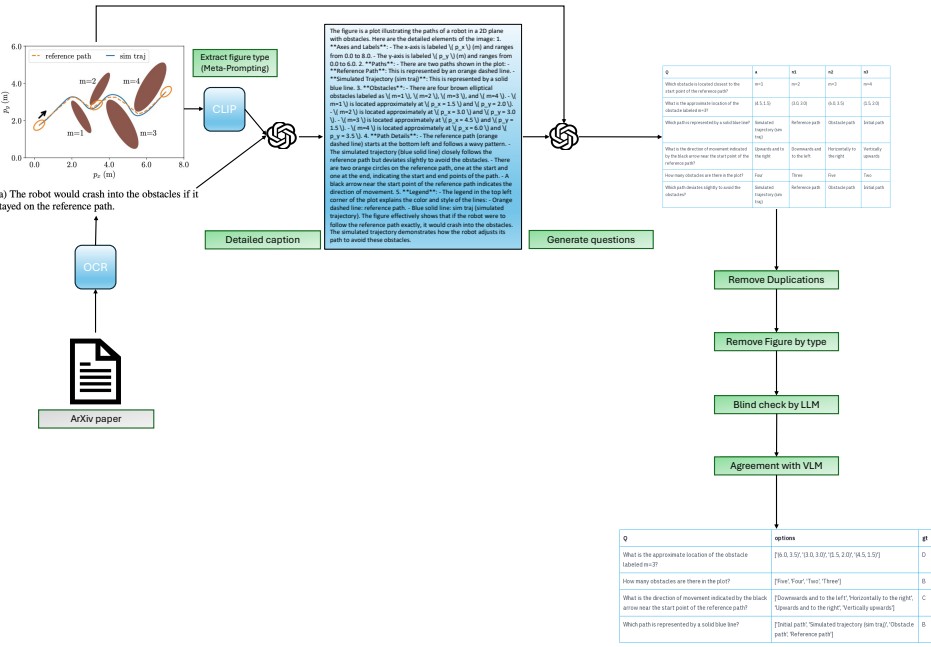

Figure 11: A detailed example for VQA questions generation.

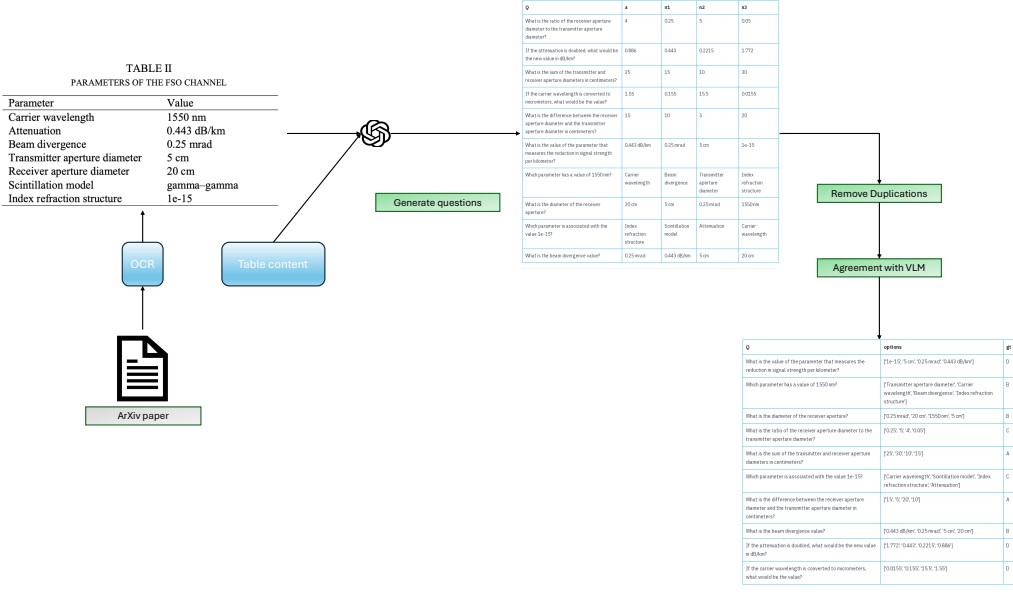

Figure 12: A detailed example for TQA question generation.

Table 11: Version 0 - VQA and TQA average accuracy across ArXiv taxonomy (the number of samples is in brackets).

| VQA Accuracy | cs.AI (188) | cs.CV (200) | cs.LG (196) | cs.RO (321) | eess.SP (269) | eess.SY (286) | q-bio.BM (154) | q-bio.CB (206) | q-bio.GN (80) | q-bio.TO (167) | Mean (2067) |
|---|---|---|---|---|---|---|---|---|---|---|---|
| InstrcutBLIP | 0.29 | 0.24 | 0.20 | 0.22 | 0.24 | 0.25 | 0.27 | 0.22 | 0.19 | 0.25 | 0.24 |
| InterLM-XC-2.5 | 0.42 | 0.42 | 0.49 | 0.48 | 0.56 | 0.48 | 0.51 | 0.49 | 0.57 | 0.46 | 0.49 |
| InternVL2-2B | 0.30 | 0.28 | 0.30 | 0.34 | 0.32 | 0.36 | 0.26 | 0.25 | 0.22 | 0.30 | 0.29 |
| InternVL2-8B | 0.61 | 0.66 | 0.64 | 0.65 | 0.69 | 0.68 | 0.58 | 0.55 | 0.59 | 0.57 | 0.62 |
| LLaVA1.6-7B | 0.54 | 0.56 | 0.52 | 0.62 | 0.55 | 0.60 | 0.50 | 0.50 | 0.60 | 0.43 | 0.54 |
| LLaVA-OneVision | 0.32 | 0.36 | 0.33 | 0.38 | 0.35 | 0.35 | 0.31 | 0.27 | 0.32 | 0.22 | 0.32 |
| LLaVA1.5-13B | 0.29 | 0.35 | 0.31 | 0.35 | 0.33 | 0.33 | 0.33 | 0.28 | 0.29 | 0.25 | 0.31 |
| LLaVA1.5-7B | 0.44 | 0.50 | 0.63 | 0.52 | 0.52 | 0.53 | 0.46 | 0.40 | 0.44 | 0.48 | 0.49 |
| LLaVA1.6-34B | 0.30 | 0.39 | 0.33 | 0.36 | 0.29 | 0.36 | 0.33 | 0.32 | 0.32 | 0.31 | 0.33 |
| Mantis | 0.55 | 0.60 | 0.65 | 0.57 | 0.67 | 0.61 | 0.54 | 0.58 | 0.57 | 0.42 | 0.58 |
| Phi3v | 0.42 | 0.42 | 0.41 | 0.41 | 0.29 | 0.35 | 0.39 | 0.36 | 0.39 | 0.26 | 0.37 |
| InterLM-XC-4Khd | 0.39 | 0.44 | 0.42 | 0.41 | 0.43 | 0.45 | 0.38 | 0.34 | 0.32 | 0.40 | 0.40 |
| Idefics2 | 0.47 | 0.48 | 0.48 | 0.52 | 0.50 | 0.49 | 0.49 | 0.43 | 0.49 | 0.48 | 0.48 |
| Claude-Sonnet | 0.77 | 0.74 | 0.81 | 0.77 | 0.89 | 0.79 | 0.72 | 0.74 | 0.55 | 0.78 | 0.75 |
| Qwen2-VL-7B | 0.62 | 0.66 | 0.71 | 0.72 | 0.77 | 0.70 | 0.69 | 0.59 | 0.66 | 0.61 | 0.67 |
| GPT4o | 0.49 | 0.52 | 0.55 | 0.55 | 0.58 | 0.52 | 0.47 | 0.49 | 0.45 | 0.49 | 0.51 |
| Idefics3 | 0.52 | 0.63 | 0.54 | 0.58 | 0.54 | 0.54 | 0.55 | 0.47 | 0.55 | 0.51 | 0.54 |

| TQA Accuracy | cs.AI (167) | cs.CV (473) | cs.LG (516) | cs.RO (247) | eess.SP (45) | eess.SY (188) | q-bio.BM (118) | q-bio.CB (108) | q-bio.GN (398) | q-bio.TO (115) | Mean (2375) |
|---|---|---|---|---|---|---|---|---|---|---|---|
| InstrcutBLIP | 0.20 | 0.19 | 0.23 | 0.19 | 0.20 | 0.21 | 0.19 | 0.14 | 0.24 | 0.21 | 0.20 |
| InterLM-XC-2.5 | 0.46 | 0.38 | 0.36 | 0.51 | 0.38 | 0.56 | 0.45 | 0.44 | 0.47 | 0.44 | 0.45 |
| InternVL2-2B | 0.41 | 0.36 | 0.36 | 0.51 | 0.27 | 0.49 | 0.42 | 0.43 | 0.40 | 0.42 | 0.41 |
| InternVL2-8B | 0.50 | 0.56 | 0.55 | 0.68 | 0.64 | 0.75 | 0.62 | 0.64 | 0.54 | 0.73 | 0.62 |
| LLaVA1.6-7B | 0.26 | 0.19 | 0.22 | 0.29 | 0.24 | 0.26 | 0.20 | 0.28 | 0.22 | 0.30 | 0.25 |
| LLaVA-OneVision | 0.48 | 0.41 | 0.43 | 0.57 | 0.33 | 0.64 | 0.49 | 0.57 | 0.43 | 0.55 | 0.49 |
| LLaVA1.5-13B | 0.37 | 0.29 | 0.28 | 0.36 | 0.31 | 0.32 | 0.29 | 0.31 | 0.30 | 0.31 | 0.31 |
| LLaVA1.5-7B | 0.32 | 0.26 | 0.29 | 0.32 | 0.33 | 0.30 | 0.28 | 0.36 | 0.30 | 0.30 | 0.31 |
| LLaVA1.6-34B | 0.54 | 0.50 | 0.46 | 0.55 | 0.47 | 0.63 | 0.58 | 0.57 | 0.47 | 0.54 | 0.53 |
| Mantis | 0.31 | 0.24 | 0.29 | 0.35 | 0.33 | 0.32 | 0.19 | 0.32 | 0.32 | 0.32 | 0.30 |
| Phi3v | 0.47 | 0.41 | 0.43 | 0.51 | 0.49 | 0.60 | 0.48 | 0.56 | 0.46 | 0.48 | 0.49 |
| InterLM-XC-4Khd | 0.43 | 0.31 | 0.36 | 0.50 | 0.24 | 0.53 | 0.48 | 0.44 | 0.38 | 0.43 | 0.41 |
| Idefics2 | 0.36 | 0.30 | 0.34 | 0.38 | 0.33 | 0.43 | 0.34 | 0.41 | 0.34 | 0.30 | 0.35 |
| Claude-Sonnet | 0.77 | 0.79 | 0.84 | 0.85 | 0.80 | 0.85 | 0.84 | 0.79 | 0.78 | 0.83 | 0.81 |
| Qwen2-VL-7B | 0.59 | 0.53 | 0.52 | 0.63 | 0.53 | 0.64 | 0.53 | 0.66 | 0.53 | 0.63 | 0.58 |
| GPT4o | 0.46 | 0.41 | 0.40 | 0.55 | 0.47 | 0.57 | 0.49 | 0.50 | 0.45 | 0.48 | 0.48 |
| Idefics3 | 0.49 | 0.43 | 0.44 | 0.57 | 0.36 | 0.57 | 0.45 | 0.58 | 0.46 | 0.52 | 0.49 |

## A.4 ADDITIONAL LIVEXIV DATASET VERSIONS

In this section we describe in detail the additional LiveXiv versions, v0 (generated form papers from 2010), v1-opposite which is the same raw data as v1 but this time Claude is the QA generation model and GPT-4o does the filtering, and v2 and v3 which consist of multiple participating models. We start by describing each version and the performance of the evaluated models. Lastly, we summarize the diversity of our generated dataset, both in terms of visual content and in terms of the different categories of questions.

### A.4.1 LIVEXIV V0

We analyzed 100 scientific papers from 2010 and developed v0, which contains 4,500 questions. Due to the dynamic nature of scientific data, v0 represents a significant distribution shift from v1. We used our efficient evaluation method to predict v1 from v0, and our high-accuracy predictions demonstrate the method's robustness across a large temporal gap. Refer to Table 11 for performance details.

### A.4.2 LIVEXIV V1-OPPOSITE

To show the impact of the the role of each model in the dataset creation, we created an opposite version of v1, where Claude is the model that generates the questions and GPT is the model that does the filtering. Table 12 demonstrates the changes in the absolute performance and in the ranking between the models. Overall, switching roles between Claude and GPT-4o has minimal impact on most models. However, GPT-4o shows slightly more variation, dropping slightly in the rankings when it takes on the role of filtering questions. This variation occurs because generating the questions provides some advantage.

### A.4.3 LIVEXIV V2

LiveXiv second version (v2) consists of 10.4K questions for VQA and 7.7K questions for TQA. In addition we introduced 4 new domains from physics, namely, physics.app-ph, physics.optics,

Table 12: We compared LiveXiv first version with different configurations, The original configuration where GPT4o generates the questions and Claude filters and an opposite version where Claude generates the questions and GPT4o performs the filtering. We see in it that overall, the ranking remains similar and the changes are minor. Indeed, for the generating model there is some advantage as we see that GPT goes down a bit in the ranking (Claude remains first in both cases).

| | VQA | | TQA | |
|---|---|---|---|---|
| | Ranking change | Average change (%) | Ranking change | Average change (%) |
| InstructBLIP-7B | 0 | -1.78 | 0 | 9.66 |
| IXC2.5-7B | 0 | -0.83 | 0 | 13.73 |
| InternVL2-8B | 0 | -1.18 | 0 | 2.37 |
| LLaVA-1.6-Mistral-7B | 2 | -8.64 | 0 | 10.72 |
| LLaVA-OneVision-7B | 1 | -3.66 | -2 | 13.42 |
| LLaVA-v1.5-13B | 0 | -6.49 | 0 | 8.48 |
| LLaVA-v1.5-7B | 0 | -8.00 | -1 | 12.72 |
| LLaVA-v1.6-34B | 2 | -5.85 | 0 | -0.35 |
| Mantis-LLama3-8B | 2 | -7.13 | 0 | 10.24 |
| Phi3v | 1 | -0.49 | 0 | 13.856 |
| IXC2-4KHD-7B | -5 | -0.47 | 3 | 8.60 |
| Idefics2-8B | 1 | -6.00 | 0 | 3.21 |
| InternVL2-2B | 0 | -1.42 | 0 | 2.61 |
| Claude-Sonnet | 0 | -3.31 | 0 | 10.11 |
| Qwen2-VL | 0 | -0.31 | 0 | 2.81 |
| GPT-4o | -4 | 4.74 | 0 | 13.46 |
| Idefics3 | 0 | -1.89 | 0 | 12.23 |
| Average | 0 | -3.10 | 0 | 8.70 |

Table 13: Version 2 - VQA and TQA average accuracy across ArXiv taxonomy (the number of samples is in brackets).

| VQA Accuracy | cs.AI (790) | cs.CV (1071) | cs.LG (853) | cs.RO (834) | eess.SP (889) | eess.SY (772) | physics.app-ph (944) | physics.bio-ph (868) | physics.data-an (841) | physics.optics (819) | q-bio.BM (890) | q-bio.CB (184) | q-bio.GN (321) | q-bio.TO (299) | Mean (10375) |
|---|---|---|---|---|---|---|---|---|---|---|---|---|---|---|---|
| InstrcutBLIP | 0.25 | 0.24 | 0.23 | 0.27 | 0.28 | 0.22 | 0.22 | 0.25 | 0.27 | 0.22 | 0.20 | 0.25 | 0.22 | 0.17 | 0.24 |
| InterLM-XC-2.5 | 0.51 | 0.49 | 0.43 | 0.49 | 0.54 | 0.52 | 0.46 | 0.52 | 0.49 | 0.51 | 0.47 | 0.42 | 0.54 | 0.51 | 0.49 |
| InternVL2-8B | 0.66 | 0.67 | 0.54 | 0.65 | 0.65 | 0.65 | 0.63 | 0.68 | 0.64 | 0.66 | 0.62 | 0.59 | 0.64 | 0.66 | 0.64 |
| InternVL2-2B | 0.34 | 0.39 | 0.34 | 0.36 | 0.33 | 0.32 | 0.30 | 0.34 | 0.32 | 0.31 | 0.34 | 0.29 | 0.30 | 0.34 | 0.33 |
| LLaVA1.6-7B | 0.55 | 0.60 | 0.47 | 0.59 | 0.54 | 0.59 | 0.57 | 0.59 | 0.55 | 0.59 | 0.55 | 0.47 | 0.58 | 0.56 | 0.56 |
| LLaVA-OneVision | 0.35 | 0.36 | 0.33 | 0.39 | 0.35 | 0.36 | 0.34 | 0.35 | 0.33 | 0.34 | 0.36 | 0.33 | 0.35 | 0.35 | 0.35 |
| LLaVA1.5-13B | 0.32 | 0.35 | 0.33 | 0.35 | 0.31 | 0.34 | 0.31 | 0.34 | 0.35 | 0.35 | 0.33 | 0.35 | 0.32 | 0.35 | 0.33 |
| LLaVA1.5-7B | 0.53 | 0.56 | 0.42 | 0.57 | 0.54 | 0.54 | 0.48 | 0.58 | 0.45 | 0.55 | 0.51 | 0.47 | 0.45 | 0.56 | 0.52 |
| LLaVA1.6-34B | 0.35 | 0.35 | 0.29 | 0.35 | 0.30 | 0.33 | 0.31 | 0.35 | 0.31 | 0.36 | 0.33 | 0.35 | 0.33 | 0.35 | 0.33 |
| Mantis | 0.59 | 0.64 | 0.51 | 0.63 | 0.64 | 0.60 | 0.60 | 0.66 | 0.60 | 0.62 | 0.60 | 0.49 | 0.58 | 0.62 | 0.60 |
| Phi3v | 0.43 | 0.45 | 0.30 | 0.42 | 0.32 | 0.35 | 0.32 | 0.33 | 0.36 | 0.32 | 0.36 | 0.24 | 0.38 | 0.32 | 0.35 |
| InterLM-XC-4khd | 0.39 | 0.44 | 0.36 | 0.43 | 0.36 | 0.40 | 0.38 | 0.43 | 0.39 | 0.39 | 0.39 | 0.33 | 0.34 | 0.39 | 0.39 |
| Idefics2 | 0.50 | 0.54 | 0.45 | 0.52 | 0.47 | 0.52 | 0.52 | 0.55 | 0.51 | 0.53 | 0.48 | 0.43 | 0.48 | 0.51 | 0.50 |
| Claude-Sonnet | 0.81 | 0.78 | 0.75 | 0.76 | 0.83 | 0.80 | 0.78 | 0.80 | 0.76 | 0.80 | 0.77 | 0.69 | 0.76 | 0.78 | 0.78 |
| Qwen2-VL-7B | 0.68 | 0.71 | 0.61 | 0.68 | 0.72 | 0.66 | 0.70 | 0.75 | 0.69 | 0.73 | 0.67 | 0.67 | 0.69 | 0.71 | 0.68 |
| GPT4o | 0.58 | 0.61 | 0.50 | 0.60 | 0.62 | 0.57 | 0.61 | 0.63 | 0.59 | 0.61 | 0.59 | 0.53 | 0.54 | 0.67 | 0.59 |
| Idefics3 | 0.57 | 0.60 | 0.50 | 0.55 | 0.58 | 0.58 | 0.52 | 0.59 | 0.53 | 0.54 | 0.55 | 0.52 | 0.60 | 0.58 | 0.56 |
| Pixtral | 0.73 | 0.71 | 0.66 | 0.73 | 0.78 | 0.70 | 0.76 | 0.77 | 0.73 | 0.73 | 0.70 | 0.67 | 0.72 | 0.77 | 0.73 |
| Molmo | 0.62 | 0.67 | 0.52 | 0.62 | 0.62 | 0.62 | 0.59 | 0.65 | 0.61 | 0.63 | 0.61 | 0.49 | 0.55 | 0.67 | 0.60 |

| TQA Accuracy | cs.AI (1208) | cs.CV (901) | cs.LG (785) | cs.RO (1315) | eess.SP (164) | eess.SY (458) | physics.app-ph (998) | physics.bio-ph (482) | physics.data-an (340) | physics.optics (198) | q-bio.BM (252) | q-bio.CB (203) | q-bio.GN (141) | q-bio.TO (267) | Mean (7712) |
|---|---|---|---|---|---|---|---|---|---|---|---|---|---|---|---|
| InstructBLIP | 0.20 | 0.22 | 0.21 | 0.19 | 0.19 | 0.19 | 0.15 | 0.13 | 0.18 | 0.18 | 0.18 | 0.21 | 0.20 | 0.18 | 0.19 |
| InterLM-XC-2.5 | 0.41 | 0.43 | 0.45 | 0.49 | 0.42 | 0.50 | 0.46 | 0.56 | 0.52 | 0.43 | 0.45 | 0.38 | 0.46 | 0.41 | 0.46 |
| InternVL2-2B | 0.37 | 0.34 | 0.39 | 0.37 | 0.36 | 0.41 | 0.41 | 0.47 | 0.39 | 0.37 | 0.38 | 0.34 | 0.41 | 0.30 | 0.38 |
| InternVL2-8B | 0.56 | 0.55 | 0.59 | 0.64 | 0.63 | 0.65 | 0.65 | 0.69 | 0.56 | 0.60 | 0.61 | 0.51 | 0.57 | 0.57 | 0.60 |
| LLaVA1.6-7B | 0.23 | 0.21 | 0.22 | 0.23 | 0.21 | 0.24 | 0.22 | 0.21 | 0.26 | 0.25 | 0.24 | 0.15 | 0.24 | 0.21 | 0.22 |
| LLaVA-OneVision | 0.42 | 0.44 | 0.50 | 0.48 | 0.45 | 0.53 | 0.49 | 0.56 | 0.52 | 0.50 | 0.48 | 0.46 | 0.49 | 0.47 | 0.49 |
| LLaVA1.5-13B | 0.30 | 0.28 | 0.33 | 0.32 | 0.34 | 0.28 | 0.29 | 0.35 | 0.31 | 0.31 | 0.29 | 0.25 | 0.28 | 0.27 | 0.30 |
| LLaVA1.5-7B | 0.28 | 0.26 | 0.31 | 0.31 | 0.28 | 0.30 | 0.27 | 0.33 | 0.28 | 0.29 | 0.31 | 0.33 | 0.25 | 0.27 | 0.29 |
| LLaVA1.6-34B | 0.45 | 0.45 | 0.48 | 0.54 | 0.54 | 0.59 | 0.55 | 0.55 | 0.55 | 0.52 | 0.54 | 0.48 | 0.49 | 0.51 | 0.52 |
| Mantis | 0.28 | 0.28 | 0.29 | 0.32 | 0.33 | 0.28 | 0.33 | 0.33 | 0.34 | 0.25 | 0.28 | 0.22 | 0.27 | 0.24 | 0.29 |
| Phi3v | 0.43 | 0.47 | 0.48 | 0.49 | 0.42 | 0.53 | 0.47 | 0.60 | 0.54 | 0.48 | 0.47 | 0.39 | 0.49 | 0.44 | 0.48 |
| InterLM-XC-4khd | 0.37 | 0.37 | 0.40 | 0.43 | 0.34 | 0.42 | 0.40 | 0.46 | 0.44 | 0.47 | 0.39 | 0.24 | 0.41 | 0.30 | 0.39 |
| Idefics2 | 0.32 | 0.33 | 0.35 | 0.37 | 0.34 | 0.44 | 0.35 | 0.44 | 0.39 | 0.36 | 0.36 | 0.29 | 0.37 | 0.32 | 0.36 |
| Claude-Sonnet | 0.79 | 0.82 | 0.81 | 0.87 | 0.81 | 0.81 | 0.83 | 0.85 | 0.78 | 0.80 | 0.83 | 0.77 | 0.81 | 0.84 | 0.82 |
| Qwen2-VL-7B | 0.51 | 0.54 | 0.56 | 0.59 | 0.60 | 0.63 | 0.57 | 0.73 | 0.59 | 0.53 | 0.56 | 0.51 | 0.61 | 0.52 | 0.58 |
| GPT4o | 0.47 | 0.45 | 0.45 | 0.50 | 0.50 | 0.50 | 0.52 | 0.55 | 0.46 | 0.46 | 0.47 | 0.43 | 0.49 | 0.54 | 0.49 |
| Idefics3 | 0.43 | 0.48 | 0.47 | 0.47 | 0.47 | 0.58 | 0.47 | 0.55 | 0.49 | 0.48 | 0.47 | 0.39 | 0.50 | 0.49 | 0.48 |
| Pixtral | 0.57 | 0.56 | 0.58 | 0.59 | 0.57 | 0.63 | 0.60 | 0.66 | 0.58 | 0.57 | 0.58 | 0.54 | 0.61 | 0.57 | 0.59 |
| Molmo | 0.49 | 0.48 | 0.51 | 0.53 | 0.49 | 0.55 | 0.52 | 0.64 | 0.53 | 0.50 | 0.53 | 0.44 | 0.51 | 0.49 | 0.51 |

physics.bio-ph, physics.data-an to a total of 14 domains. Lastly, we also added Two new models, Pixtral (Agrawal et al., 2024) and Molmo-7B (Deitke et al., 2024). Due to our high frequency updates, Quantitative Biology related domains have overleaping papers between versions. Thus, we filtered these overleaping papers out. V2 shows similar diversity to v1 (see Table 18), similar performance of the evaluated models, resulting an accurate and reliable efficient evaluation (see Figure ??) and lastly, demonstrating LiveXiv to be evolving scalable live dataset. Table 13 reports the performance summary.

## A.4.4 LiveXiv v3

To further mitigate the potential bias due to single generation and single filtering model, v3 is a combination of 3 participant models, GPT-4o, Claude-Sonnet and Qwen2-VL-72B. v3 is actually 3 subsets, where each time a different model takes a different role. Table 14 presents the results for all 3 subsets. Table 15 shows the performance per subset. Overall we can see that the model ranking remains similar to previous version. This hints that the bias of using a single model is small. We extended our manual verification process with an additional check of v3. Tables 16 and 17 present these results in detail. The manual check revealed more variations compared to the manual check on v1, which can be attributed to the participation of three models in v3. As shown in Table 15, Qwen2-VL-72B demonstrated weaker performance compared to GPT and Claude. Consequently, its role as the filtering and generation model likely introduced additional noise into the dataset. However, the average performance change for both VQA and TQA remains below 5%.

Table 14: Version 3 - VQA and TQA average accuracy across ArXiv taxonomy (the number of samples is in brackets).

| VQA Accuracy | cs.AI (609) | cs.CV (892) | cs.LG (792) | cs.RO (798) | eess.SP (835) | eess.SY (597) | physics.app-ph (1035) | physics.bio-ph (908) | physics.data-an (425) | physics.optics (628) | q-bio.BM (803) | q-bio.CB (640) | q-bio.GN (425) | q-bio.TO (555) | Mean (9942) |
|---|---|---|---|---|---|---|---|---|---|---|---|---|---|---|---|
| InstrcutBLIP | 0.24 | 0.26 | 0.23 | 0.26 | 0.21 | 0.21 | 0.21 | 0.22 | 0.22 | 0.20 | 0.22 | 0.23 | 0.26 | 0.24 | 0.23 |
| InterLM-XC-2.5 | 0.50 | 0.50 | 0.46 | 0.49 | 0.48 | 0.47 | 0.53 | 0.50 | 0.50 | 0.52 | 0.49 | 0.48 | 0.51 | 0.52 | 0.50 |
| InternVL2-2B | 0.37 | 0.38 | 0.31 | 0.42 | 0.33 | 0.35 | 0.34 | 0.31 | 0.28 | 0.34 | 0.33 | 0.35 | 0.32 | 0.35 | 0.34 |
| InternVL2-8B | 0.65 | 0.66 | 0.60 | 0.71 | 0.67 | 0.65 | 0.69 | 0.64 | 0.62 | 0.68 | 0.62 | 0.64 | 0.61 | 0.69 | 0.65 |
| LLaVA1.6-7B | 0.58 | 0.59 | 0.51 | 0.65 | 0.57 | 0.55 | 0.61 | 0.57 | 0.52 | 0.58 | 0.51 | 0.56 | 0.52 | 0.57 | 0.56 |
| LLaVA-OneVision | 0.38 | 0.40 | 0.33 | 0.45 | 0.33 | 0.39 | 0.36 | 0.34 | 0.31 | 0.37 | 0.36 | 0.38 | 0.33 | 0.34 | 0.36 |
| LLaVA1.5-13B | 0.37 | 0.38 | 0.32 | 0.44 | 0.31 | 0.36 | 0.35 | 0.33 | 0.29 | 0.36 | 0.32 | 0.36 | 0.34 | 0.36 | 0.35 |
| LLaVA1.5-7B | 0.54 | 0.59 | 0.47 | 0.64 | 0.56 | 0.56 | 0.57 | 0.52 | 0.47 | 0.59 | 0.50 | 0.49 | 0.50 | 0.57 | 0.54 |
| LLaVA1.6-34B | 0.37 | 0.40 | 0.31 | 0.42 | 0.33 | 0.37 | 0.37 | 0.35 | 0.30 | 0.36 | 0.34 | 0.37 | 0.34 | 0.37 | 0.36 |
| Mantis | 0.63 | 0.64 | 0.59 | 0.65 | 0.62 | 0.59 | 0.67 | 0.63 | 0.60 | 0.63 | 0.56 | 0.60 | 0.60 | 0.59 | 0.61 |
| Phi3v | 0.38 | 0.42 | 0.34 | 0.46 | 0.33 | 0.33 | 0.38 | 0.32 | 0.31 | 0.37 | 0.31 | 0.31 | 0.37 | 0.38 | 0.36 |
| InterLM-XC-4Khd | 0.40 | 0.42 | 0.35 | 0.47 | 0.36 | 0.41 | 0.42 | 0.39 | 0.35 | 0.38 | 0.36 | 0.39 | 0.39 | 0.45 | 0.40 |
| Idefics2 | 0.54 | 0.55 | 0.47 | 0.58 | 0.49 | 0.51 | 0.55 | 0.50 | 0.49 | 0.50 | 0.48 | 0.53 | 0.49 | 0.50 | 0.51 |
| Claude-Sonnet | 0.75 | 0.77 | 0.76 | 0.78 | 0.80 | 0.80 | 0.81 | 0.80 | 0.73 | 0.81 | 0.76 | 0.75 | 0.77 | 0.78 | 0.78 |
| Qwen2-VL-7B | 0.68 | 0.69 | 0.66 | 0.74 | 0.68 | 0.66 | 0.75 | 0.72 | 0.69 | 0.72 | 0.62 | 0.70 | 0.68 | 0.73 | 0.69 |
| GPT4o | 0.50 | 0.54 | 0.50 | 0.56 | 0.51 | 0.54 | 0.52 | 0.48 | 0.48 | 0.48 | 0.46 | 0.51 | 0.51 | 0.52 | 0.51 |
| Idefics3 | 0.59 | 0.58 | 0.50 | 0.60 | 0.54 | 0.57 | 0.58 | 0.53 | 0.52 | 0.56 | 0.49 | 0.52 | 0.57 | 0.54 | 0.55 |
| Pixtral | 0.69 | 0.71 | 0.69 | 0.74 | 0.72 | 0.70 | 0.76 | 0.70 | 0.72 | 0.74 | 0.69 | 0.71 | 0.70 | 0.74 | 0.71 |
| Molmo | 0.62 | 0.64 | 0.58 | 0.66 | 0.64 | 0.60 | 0.66 | 0.60 | 0.58 | 0.62 | 0.57 | 0.62 | 0.60 | 0.65 | 0.62 |

| TQA Accuracy | cs.AI (802) | cs.CV (939) | cs.LG (890) | cs.RO (502) | eess.SP (386) | eess.SY (273) | physics.app-ph (270) | physics.bio-ph (199) | physics.data-an (149) | physics.optics (241) | q-bio.BM (821) | q-bio.CB (283) | q-bio.GN (650) | q-bio.TO (248) | Mean (6653) |
|---|---|---|---|---|---|---|---|---|---|---|---|---|---|---|---|
| InstrcutBLIP | 0.18 | 0.25 | 0.16 | 0.18 | 0.17 | 0.22 | 0.24 | 0.18 | 0.18 | 0.19 | 0.22 | 0.18 | 0.20 | 0.23 | 0.20 |
| InterLM-XC-2.5 | 0.49 | 0.47 | 0.51 | 0.49 | 0.49 | 0.59 | 0.66 | 0.65 | 0.52 | 0.53 | 0.50 | 0.56 | 0.56 | 0.59 | 0.54 |
| InternVL2-2B | 0.44 | 0.38 | 0.44 | 0.46 | 0.46 | 0.56 | 0.63 | 0.58 | 0.46 | 0.46 | 0.42 | 0.49 | 0.47 | 0.47 | 0.48 |
| InternVL2-8B | 0.67 | 0.60 | 0.60 | 0.65 | 0.67 | 0.77 | 0.79 | 0.77 | 0.70 | 0.72 | 0.62 | 0.75 | 0.66 | 0.67 | 0.69 |
| LLaVA1.6-7B | 0.26 | 0.24 | 0.24 | 0.21 | 0.23 | 0.32 | 0.35 | 0.29 | 0.29 | 0.26 | 0.24 | 0.27 | 0.23 | 0.26 | 0.26 |
| LLaVA-OneVision | 0.52 | 0.52 | 0.50 | 0.52 | 0.52 | 0.66 | 0.73 | 0.66 | 0.60 | 0.57 | 0.48 | 0.62 | 0.53 | 0.56 | 0.57 |
| LLaVA1.5-13B | 0.35 | 0.32 | 0.33 | 0.29 | 0.34 | 0.38 | 0.33 | 0.32 | 0.17 | 0.35 | 0.33 | 0.29 | 0.33 | 0.24 | 0.31 |
| LLaVA1.5-7B | 0.32 | 0.32 | 0.32 | 0.30 | 0.34 | 0.37 | 0.33 | 0.30 | 0.24 | 0.30 | 0.32 | 0.31 | 0.30 | 0.29 | 0.31 |
| LLaVA1.6-34B | 0.58 | 0.51 | 0.47 | 0.56 | 0.60 | 0.66 | 0.72 | 0.72 | 0.60 | 0.65 | 0.55 | 0.67 | 0.56 | 0.54 | 0.60 |
| Mantis | 0.32 | 0.27 | 0.33 | 0.31 | 0.29 | 0.40 | 0.33 | 0.32 | 0.33 | 0.35 | 0.29 | 0.30 | 0.30 | 0.26 | 0.31 |
| Phi3v | 0.52 | 0.51 | 0.53 | 0.51 | 0.55 | 0.62 | 0.74 | 0.68 | 0.59 | 0.61 | 0.53 | 0.63 | 0.58 | 0.51 | 0.58 |
| InterLM-XC-4Khd | 0.45 | 0.39 | 0.46 | 0.44 | 0.44 | 0.51 | 0.57 | 0.55 | 0.43 | 0.48 | 0.46 | 0.46 | 0.44 | 0.47 | 0.47 |
| Idefics2 | 0.40 | 0.31 | 0.39 | 0.33 | 0.40 | 0.46 | 0.51 | 0.45 | 0.42 | 0.47 | 0.35 | 0.42 | 0.42 | 0.44 | 0.41 |
| Claude-Sonnet | 0.82 | 0.78 | 0.78 | 0.85 | 0.85 | 0.87 | 0.85 | 0.86 | 0.89 | 0.88 | 0.78 | 0.88 | 0.83 | 0.83 | 0.84 |
| Qwen2-VL-7B | 0.62 | 0.60 | 0.63 | 0.63 | 0.62 | 0.68 | 0.77 | 0.72 | 0.72 | 0.73 | 0.62 | 0.70 | 0.66 | 0.64 | 0.67 |
| GPT4o | 0.45 | 0.44 | 0.45 | 0.44 | 0.51 | 0.59 | 0.62 | 0.56 | 0.54 | 0.53 | 0.48 | 0.60 | 0.54 | 0.51 | 0.52 |
| Idefics3 | 0.55 | 0.49 | 0.57 | 0.47 | 0.51 | 0.59 | 0.68 | 0.66 | 0.54 | 0.59 | 0.52 | 0.58 | 0.56 | 0.53 | 0.56 |
| Pixtral | 0.40 | 0.41 | 0.50 | 0.42 | 0.37 | 0.48 | 0.16 | 0.14 | 0.49 | 0.54 | 0.36 | 0.42 | 0.39 | 0.37 | 0.39 |
| Molmo | 0.35 | 0.35 | 0.42 | 0.38 | 0.34 | 0.47 | 0.14 | 0.12 | 0.44 | 0.52 | 0.31 | 0.33 | 0.34 | 0.31 | 0.34 |

Table 15: **Average performance (%) on different versions.** We clearly can see that version subset 3 (denote v3-s3) has a large shift in the average performance for TQA, suggests that Qwen2-VL-72B might generates easier questions.

| | VQA | TQA |
|---|---|---|
| v1 | 47.1 | 45.1 |
| v2 | 50.7 | 44.0 |
| v3 - s1 | 51.6 | 44.5 |
| v3 - s2 | 49.1 | 41.7 |
| v3 - s3 | 52.2 | 52.5 |
| v3 | 50.8 | 46.3 |

Table 16: VQA Manual verification. We re-verified our dataset by curating a subset of 300 questions (100 from each subset). We can see that version 3 even though composed of 3 different subset, remains accurate with a bounded performance change overall and minimal ranking changes.

| | Overall | Manual | Performance change | Rank Overall | Rank Manual | Rank change |
|---|---|---|---|---|---|---|
| instructblip-vicuna-7b | 22.90 | 26.75 | -3.84 | 19 | 19 | 0 |
| internlm-xcomposer2d5-7b | 49.62 | 51.37 | -1.75 | 12 | 12 | 0 |
| InternVL2-8B | 65.43 | 68.09 | -2.66 | 4 | 5 | -1 |
| llava-v1.6-mistral-7b | 34.21 | 40.12 | -5.91 | 18 | 17 | 1 |
| llava-onevision-qwen2-7b-ov | 56.69 | 59.88 | -3.19 | 7 | 9 | -2 |
| llava-v1.5-13b | 36.48 | 43.16 | -6.68 | 14 | 15 | -1 |
| llava-v1.5-7b | 34.92 | 39.51 | -4.59 | 17 | 18 | -1 |
| llava-v1.6-34b | 54.59 | 61.40 | -6.81 | 9 | 8 | 1 |
| Mantis-8B-siglip-llama3 | 35.87 | 41.34 | -5.47 | 16 | 16 | 0 |
| phi3v | 61.79 | 64.74 | -2.95 | 6 | 7 | -1 |
| internlm-xcomposer2-4khd-7b | 36.07 | 46.20 | -10.13 | 15 | 13 | 2 |
| idefics2-8b | 39.64 | 45.59 | -5.95 | 13 | 14 | -1 |
| InternVL2-2B | 51.67 | 54.10 | -2.43 | 10 | 10 | 0 |
| claude | 77.95 | 81.46 | -3.51 | 1 | 1 | 0 |
| qwen2vl | 69.56 | 73.56 | -3.99 | 3 | 2 | 1 |
| gpt | 50.82 | 51.67 | -0.85 | 11 | 11 | 0 |
| idefics3 | 54.94 | 64.74 | -9.80 | 8 | 7 | 1 |
| pixtral | 71.49 | 72.64 | -1.15 | 2 | 3 | -1 |
| molmo | 61.93 | 69.91 | -7.98 | 5 | 4 | 1 |
| mean | 50.87 | 55.59 | 4.72 | | | |

Table 17: TQA Manual verification. We re-verified our dataset by curating a subset of 300 questions (100 from each subset). We can see that version 3 even though composed of 3 different subset, remains accurate with small performance change overall and minimal ranking changes.

| | Overall | Manual | Performance change | Rank Overall | Rank Manual | Rank change |
|---|---|---|---|---|---|---|
| instructblip-vicuna-7b | 19.93 | 17.96 | 1.97 | 19 | 19 | 0 |
| internlm-xcomposer2d5-7b | 52.28 | 49.40 | 2.88 | 9 | 11 | -2 |
| InternVL2-2B | 45.69 | 43.71 | 1.98 | 12 | 13 | -1 |
| InternVL2-8B | 65.93 | 73.65 | -7.73 | 2 | 2 | 0 |
| llava-v1.6-mistral-7b | 25.22 | 26.35 | -1.13 | 18 | 18 | 0 |
| llava-onevision-qwen2-7b-ov | 53.99 | 56.29 | -2.30 | 8 | 6 | 2 |
| llava-v1.5-13b | 32.24 | 31.74 | 0.50 | 15 | 17 | -2 |
| llava-v1.5-7b | 31.50 | 33.23 | -1.73 | 16 | 15 | 1 |
| llava-v1.6-34b | 56.74 | 62.57 | -5.83 | 5 | 3 | 2 |
| Mantis-8B-siglip-llama3 | 30.74 | 32.34 | -1.60 | 17 | 16 | 1 |
| phi3v | 55.49 | 52.40 | 3.10 | 6 | 7 | -1 |
| internlm-xcomposer2-4khd-7b | 45.44 | 40.12 | 5.32 | 13 | 14 | -1 |
| idefics2-8b | 39.13 | 46.71 | -7.58 | 14 | 12 | 2 |
| claude | 81.92 | 88.62 | -6.70 | 1 | 1 | 0 |
| qwen2vl | 64.36 | 62.28 | 2.09 | 3 | 4 | -1 |
| gpt | 49.23 | 50.30 | -1.07 | 11 | 10 | 1 |
| idefics3 | 54.41 | 52.10 | 2.32 | 7 | 8 | -1 |
| pixtral | 57.45 | 56.36 | 1.09 | 4 | 5 | -1 |
| molmo | 50.12 | 51.36 | -1.25 | 10 | 9 | 1 |
| mean | 47.99 | 48.82 | 3.06 | | | |

### A.4.5 DIVERSITY THROUGH TIME

We evaluate LiveXiv diversity though time in Table 18. It shows that each version has a steady distribution of visual content and a diverse set of questions for both figures and tables.

Table 18: Questions and visual content diversity through LiveXiv versions, we can see that the diversity is persistent and LiveXiv has a rich set of visual content as well diverse questions throughout the new versions.

| VQA | Data Analysis | Reasoning | Attribute | Localization | Reading | Arithmetic | Charts | Block Diagram | Qualitative |
|---|---|---|---|---|---|---|---|---|---|
| v0 | 768 | 280 | 256 | 365 | 320 | 42 | 1239 | 578 | 214 |
| v1 | 2291 | 872 | 903 | 1596 | 1470 | 154 | 4354 | 2110 | 864 |
| v2 | 3005 | 1471 | 1610 | 1797 | 1732 | 112 | 5480 | 2805 | 1442 |
| v3 | 3265 | 1196 | 1459 | 1202 | 1916 | 321 | 5113 | 2141 | 1573 |

| TQA | Data Analysis | Reasoning | Attribute | Localization | Reading | Arithmetic | | | |
|---|---|---|---|---|---|---|---|---|---|
| v0 | 637 | 45 | 40 | 14 | 425 | 1168 | | | |
| v1 | 2582 | 123 | 121 | 23 | 2127 | 3934 | | | |
| v2 | 2289 | 228 | 74 | 23 | 1125 | 3436 | | | |
| v3 | 1808 | 130 | 97 | 16 | 1590 | 2572 | | | |

## A.5    EXPLORING MORE DETAILS ON EFFICIENT EVALUATION

### A.5.1    DETAILED RESULTS FOR LIVEXIV V4

In Figure 13, we present a more detailed set of results extracted from the experiment presented in Figures 4/5 for LiveXiv v4. Interestingly, we see that the models selected for re-evaluation cover the possible values of performances and that our method can make reasonable predictions for domains that were not included in LiveXiv v1.

### A.5.2    LIVEXIV V0 (PAPERS FROM 2010) EFFICIENT EVALUATION RESULTS

In Figure 15 we show the performance prediction results of our efficient re-evaluation method on v1 from v0, when re-evaluating 5 models. Despite the (temporal) big distribution shift, our efficient evaluation method does still work well.

### A.5.3    LIVEXIV V3 EFFICIENT EVALUATION
### RESULTS CONSIDERING ALL QUESTIONS (GENERATED BY DIFFERENT MODELS)

The third version of LiveXiv (denoted as v3) introduced multiple participating models for QA generation and filtering. Unlike the default pipeline for the two first versions, in which we have GPT4o as the generation model and Claude-Sonnet for the filtering, we augment LiveXiv v3 with more questions but (i) using Claude-Sonnet as the generation model and Qwen2-VL-72B as the filtering model, and using (ii) Qwen2-VL-72B as the generation model and GPT4o as the filtering model. The results can be seen in Figure 14; in summary, we have a greater distribution shift when it comes to TQA and when Qwen2-VL-72B is included as the generation model, making the prediction task for the IRT model harder, thus increasing the prediction errors for v3 when compared to Figures 4/5. In v4 we narrow the participants only to GPT-4o and Claude-Sonnet playing both roles, and reducing both potential bias from a single model for generation or filtering and maintaining accurate predictions.

### A.5.4    EFFICIENT EVALUATION ON MM-LIVEBENCH

In this section, we challenge our efficient evaluation method, by examining its performance over another type of multi-modal live dataset Zhang et al. (2024b). The dataset has 3 versions (May 2024, June 2024, and July 2024), and each version has roughly 250-300 samples of open-ended questions scraped from newspapers. To evaluate our method we use GPT-4o to convert the open-ended questions into closed-form of questions where the true answer is rephrased and 3 more negative answers are proposed. Then we evaluate 13 LMMs over all the dataset versions. We use the first version as a training set and we predict the performance over the new concatenated sets using our IRT-based method. Figure 16 shows that our method still performs well on a different benchmark when re-evaluating only 5 models.

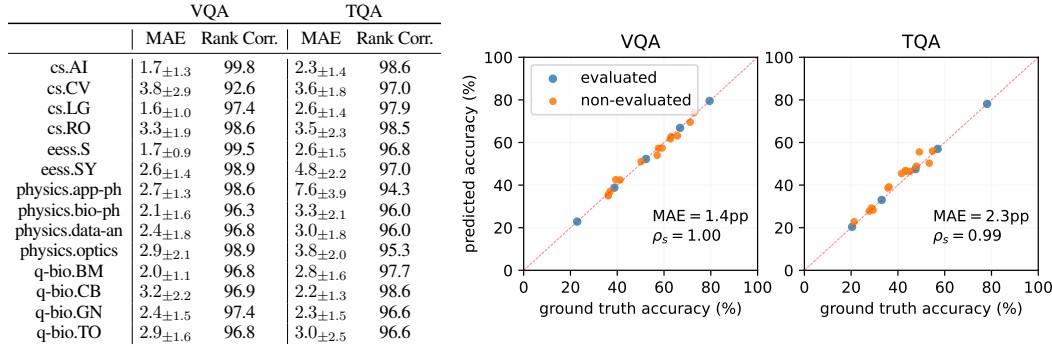

| | VQA | | TQA | |
| --- | --- | --- | --- | --- |
| | MAE | Rank Corr. | MAE | Rank Corr. |
| cs.AI | $1.7_{\pm 1.3}$ | 99.8 | $2.3_{\pm 1.4}$ | 98.6 |
| cs.CV | $3.8_{\pm 2.9}$ | 92.6 | $3.6_{\pm 1.8}$ | 97.0 |
| cs.LG | $1.6_{\pm 1.0}$ | 97.4 | $2.6_{\pm 1.4}$ | 97.9 |
| cs.RO | $3.3_{\pm 1.9}$ | 98.6 | $3.5_{\pm 2.3}$ | 98.5 |
| eess.S | $1.7_{\pm 0.9}$ | 99.5 | $2.6_{\pm 1.5}$ | 96.8 |
| eess.SY | $2.6_{\pm 1.4}$ | 98.9 | $4.8_{\pm 2.2}$ | 97.0 |
| physics.app-ph | $2.7_{\pm 1.3}$ | 98.6 | $7.6_{\pm 3.9}$ | 94.3 |
| physics.bio-ph | $2.1_{\pm 1.6}$ | 96.3 | $3.3_{\pm 2.1}$ | 96.0 |
| physics.data-an | $2.4_{\pm 1.8}$ | 96.8 | $3.0_{\pm 1.8}$ | 96.0 |
| physics.optics | $2.9_{\pm 2.1}$ | 98.9 | $3.8_{\pm 2.0}$ | 95.3 |
| q-bio.BM | $2.0_{\pm 1.1}$ | 96.8 | $2.8_{\pm 1.6}$ | 97.7 |
| q-bio.CB | $3.2_{\pm 2.2}$ | 96.9 | $2.2_{\pm 1.3}$ | 98.6 |
| q-bio.GN | $2.4_{\pm 1.5}$ | 97.4 | $2.3_{\pm 1.5}$ | 96.6 |
| q-bio.TO | $2.9_{\pm 1.6}$ | 96.8 | $3.0_{\pm 2.5}$ | 96.6 |

Figure 13: Performance prediction results of our efficient re-evaluation method on v4 when re-evaluating 5 models.

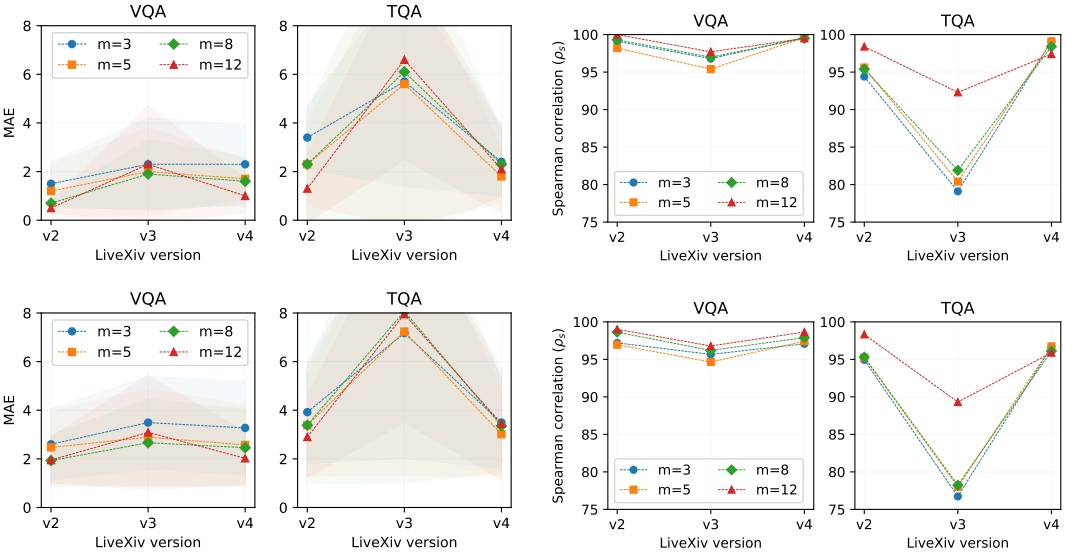

Figure 14: This figure is a reproduction of Figures 4 (first row) and 5 (second row) considering the full LiveXiv v3 and not only questions generated by GPT4o and Claude-Sonnet. In this case, we can see that the prediction performance of IRT can be deteriorated due to distribution shifts in the data.

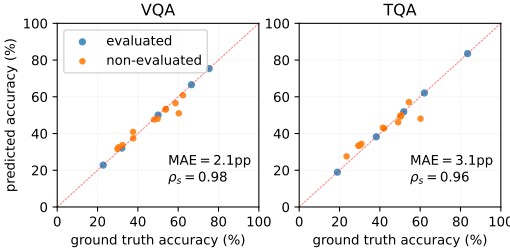

Figure 15: Performance prediction results of our efficient re-evaluation method on v1 from v0 when re-evaluating 5 models. We can see that even in a large temporal gap, our method managed to predict accurately.

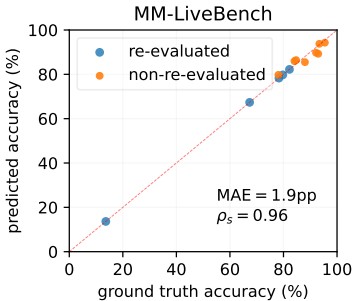

Figure 16: Our efficient evaluation method applied to MM-LiveBench. We check that our method could be successfully applied in this other context, here we convert MM-LiveBench to multiple choice questions and apply our efficient evaluation method between MM-LiveBench versions. The results for MM-LiveBench are optimistic, showing our method predict the results between its version accurately.

A.6  PROMPT TEMPLATE FOR QA GENERATIONS

```
This is a figure
 from a scientific paper with the following caption: {text_desc}.
Please describe the image in
as much details as possible. For all the details you are confident
 about include everything you see, and be as specific as possible
, such as existing numbers,  describing objects, attributes ...
```

Figure 17: Prompt template for general detailed caption.

```
Compositional reasoning defines the understanding of
 attributes, relations and word order significance. A good vision
-language model should be able to accurately answer composition
 reasoning questions about an image. Your task is to fool a vision
-language model by generating challenging compositional reasoning
 questions about the figure. Given the image and the description
 you generated: {detailed_description}, generate {n_questions}
 diverse and challenging compositional reasoning questions which a
 vision-language model would incorrectly answer. For each question
 include the following: - A compositional reasoning question -
 A correct answer - 3 hard negative options.  Each negative option
 should differ only subtly from the correct answer but still be
 clearly incorrect given the image, and the question. The goal is
for a vision-language model to choose the negative option over the
 positive option when you asked to answer the question in binary
multiple choice format.  Only include questions you are confident
in your answer and make sure there is indeed only a single correct
 answer and the others are false answers.  Format your response
 as a string in the format [{"Q":<question>, "a":<correct answer
>, "n1":<negative option 1>, "n2":<negative option 2>, ...}].
```

Figure 18: Prompt template for visual question-answering.

```
Document and table understanding defines
 the understanding of values, metrics and perform arithmetic
 operations over numerical values and commonsense reasoning
. A good language model should be able to accurately answer
{commonsense_reasoning / arithmetic manipulation} questions from a
 given table. Your task is to fool a language model by generating
challenging table {commonsense_reasoning / arithmetic manipulation
} questions about the table. Given the table: {table_content}
Generate {n_questions} diverse and challenging
{commonsense_reasoning / arithmetic manipulation} questions on the
 table questions which a language model would incorrectly answer
.For each question include the following: – A question – A correct
 answer – 3 hard negative options. Each negative option should
 differ only subtly from the correct answer but still be clearly
 incorrect given the figure, caption and the question. The goal
 is for a language model to choose the negative option over the
 positive option when you asked to answer the question in binary
 multiple choice format. Only include questions you are confident
in your answer and make sure there is indeed only a single correct
 answer and the others are false answers. Format your response
 as a string in the format [{"Q":<question>, "a":<correct answer
>, "n1":<negative option 1>, "n2":<negative option 2>, ...}].
```

Figure 19: Prompt template for table question-answering.

```
Think step by step before answering.
For the given image and question: {question}
write only the words yes or no if think the option {correct_answer
} is indeed the correct answer out of {options} for this question?
```

Figure 20: Prompt template for agreement filtering.

```
You are
 an insightful assistant, for the question/options pair provided
 by the user, pick a question category from the list below:
Question category:
- attribute: the question asks about the presence or
 visibility of an attribute of an object (e.g. "What is the color
 of circles in plot (a)?" "[A. Blue, B. White, C. Green, D. Red]")
- reasoning: the question
 asks about understanding the figure (e.g "What is the object
inside the red box?" "[A. Bottle, B. Table, C. Tree, D. Nothing]")
- localization: the question asks about
 the presence or visibility at a specific location in the image
 (e.g "On which subplot does the scatter is the most spread?"
"[A. Top-Left, B. Bottom-Right, C. 'Middle-Left', D. 'Top-Right]")
- reading: the question asks about reading
 some text from the figure (e.g "What is name of the method
presneted as a green line?" "[A. GPSK, B. FDAH, C. TQWA, D.Ours]")
- arithmetic: the questions asks about mathematical arithmetic
 of numbers (e.g if the maximum accuracy of SIFT would be
 doubled? what would be the value?" "[A. 2, B. 4 , C. 100, D. 50])
- data
 analysis: the question asks about understanding of a graph (e.g,
 "Which values intersect at T=2?" "["A. N1, B. N2, C. N3, D. N4]")
Respond with a JSON object
 with the following format: {"Question category": "category"}
```

Figure 21: Prompt template for question categories analysis.

