# OpenReview forum: "LiveXiv - A Multi-Modal live benchmark based on Arxiv papers content"
_ICLR.cc/2025/Conference — ICLR 2025 Poster_

### Official Review · Reviewer_VmiA · 2024-10-28

**Soundness:** 2
**Presentation:** 2
**Contribution:** 2
**Rating:** 5
**Confidence:** 3

**Summary:**

The study presents LiveXiv, a multi-modal live VQA benchmark designed to evaluate the capabilities of LMMs on scientific content from ArXiv papers. The benchmark addresses the issue of test data contamination by continuously evolving and utilizing the latest data, thereby providing an up-to-date assessment of model performance. LiveXiv automatically VQA pairs from figures, charts, and tables in scientific manuscripts without human intervention, leveraging advanced LMMs like GPT-4o and Claude. The study also introduces an efficient evaluation approach that estimates the performance of all models on the evolving benchmark by evaluating only a subset, thereby reducing evaluation costs.

**Strengths:**

1) It is a innovative approach to creating a live, contamination-free benchmark that is both scalable and efficient.
2) The use of advanced LMMs for automatic generation and filtering of VQA pairs is commendable, as is the development of an efficient evaluation method inspired by Item Response Theory, which reduces computational overhead.
3) he benchmark's design allows for the expansion into new domains, potentially increasing its relevance and applicability.

**Weaknesses:**

1) The scope of work presented may not be substantial enough for a benchmark paper, as the number of samples and the diversity of domains covered could be expanded.
2) While the idea of a contamination-free benchmark is valuable, the study's focus on Knowledge-VQA tasks narrows its applicability and relevance. Data contamination is a broader issue that extends beyond Knowledge-VQA tasks.
3) The reliance on proprietary LMMs for the benchmark's operation introduces potential variability and a lack of control over the evaluation process
4) The efficient evaluation method shows promise, its effectiveness and reliability over time and across different versions of the benchmark need further validation.

**Questions:**

Please address my concerns above.

---

> ### Author Response · Authors · 2024-11-21
>
> We would like to thank the reviewer for the good and important comments that helped us improve the work significantly.
> LiveXiv is a live dataset. Thus, during the review period we published in huggingface version 2 anonymously, consisting of 18K samples. We introduced 4 new domains from physics (namely, physics.optics, physics.bio-ph, physics.app-ph, physics.data-an), and added two more LMMs (Pixtral and Molmo-7B).
> We are now building version 3, which will have multiple generation and filtering models (GPT4o, Claude-Sonnet, Qwen2-VL-72B) to reduce potential biases and increase diversity. We expect to have this version published with all the analyses before the end of the discussion period.
> In addition, to perform deeper analysis, we created a variant of the first version of the dataset (that was reported in the paper) with opposite roles (Claude as the model that generates the QAs and GPT as the filter model), and validated that the overall performance and ranking remains the same.
> Lastly, we created a LiveXiv version (denoted as v0) from papers more than 10 years ago.  Our experiments on it reassure our findings reported in the paper about the performance of our efficient evaluation showing that it is also valid in the presence of a large time gap between the dataset versions.
>
> **The paper's benchmark scope appears limited, with potential for expansion in sample size and domain diversity.**
> LiveXiv is a flexible method that supports various input sources. For version 2 we extended our domains to be 14 (adding 4 additional domains from physics).
> LiveXiv can accept any arbitrary source of pdf files that we can increase and adapt the diversity as needed.
> At each version LiveXiv has over 16K questions generated from 4-5.5K images which is on par with all well established scientific datasets:
> ChartQA dataset - 33K questions (for all sets, while the test-set has only 3K questions).
> DocVQA dataset - 5K images and 50K images.
> AI2D dataset - 5K images and 4.5K questions.
> Overall after two versions of LiveXiv our dataset is larger than the leading scientific multi-modal benchmarks.
>
> **The study's narrow focus on Knowledge-VQA tasks limits its broader applicability in addressing data contamination across different domains.**
>
> While our proposed benchmark reduces contamination, we are focused on efficient generation and evaluation.
> We are motivated by using a rapidly and unified growing and adapting data source that can hint at contamination issues.
> Indeed, we demonstrate our approach on VQA, yet, our approach can be extended beyond it."
>
> **The reliance on proprietary LMMs for the benchmark's operation introduces potential variability and a lack of control over the evaluation process**
>
> We analyze the diversity of our dataset by the textual content in Section 4.3, and Tables 8,9. Our analysis shows that LiveXiv is diverse in terms of the different questions categories for both tasks.
> In addition, we will add the difficulty distribution of the question to show how LiveXiv is indeed a challenging dataset for LMMs.
> We also add a comparison to the first version where we switched the rules and now GPT4o is the filtering model and analyzes the effect of each rule. We can see that overall the relative ranking and performance remains similar.
> Moreover, we created v0 from papers 10 years ago, and reassured our evaluation is steady across time where the efficient evaluation estimates accurately (correlation 0.97 on average and MAE 2.7pp)
>
> **The efficient evaluation method shows promise, its effectiveness and reliability over time and across different versions of the benchmark need further validation.**
>
> We demonstrate the robustness of our efficient evaluation method in several scenarios. In the paper we demonstrated the effectiveness of our efficient evaluation method by splitting LiveXiv first version.
> We already verify the accuracy of the estimation of predicting models in version 2 from version 1, even on new domains introduced in LiveXiv version 2 (we will update Figure 4).
> We further stress our method by generating version 0 - a version of LiveXiv based on papers from 2010. We also find that our method is robust to predict the v1 from v0 (correlation of 0.97 and MAE of 2.7pp on average).
> In addition, we also converted the existing multi-modal live bench presented in [1] into a closed set of QA and demonstrated that our method can be applied to other live datasets as well See Section A.3.2 in the appendix for more details.
>
> [1] LMMs-Eval: Reality Check on the Evaluation of Large Multimodal Models, Zhang, Kaichen and Li, Bo and Zhang, Peiyuan and Pu, Fanyi and Cahyono, Joshua Adrian and Hu, Kairui and Liu, Shuai and Zhang, Yuanhan and Yang, Jingkang and Li, Chunyuan and others, arXiv preprint arXiv:2407.12772, 2024.
>
> \
> \
> If we have adequately addressed all your concerns, we kindly ask you to reconsider your rating. If there are any additional questions or issues, we would be happy to provide further clarification.

---

> > ### Author Response · Authors · 2024-12-02
> >
> > Dear Reviewer VmiA,
> >
> > We greatly appreciate your detailed feedback on our work, which has been valuable in improving its quality. All your comments have been thoroughly addressed in the revised version. We kindly request you to consider updating your rating if the revisions align with your expectations, and we would be glad to assist with any further questions or concerns you may have.
> >
> > Warm regards,

---

### Official Review · Reviewer_LXM8 · 2024-11-01

**Soundness:** 3
**Presentation:** 3
**Contribution:** 3
**Rating:** 6
**Confidence:** 3

**Summary:**

This paper introduces an evolving benchmark derived from arXiv papers, utilizing LLMs to automatically identify and generate visual-based question-answer pairs from the multi-modal content within manuscripts. To manage the growing dataset efficiently, an evaluation method is proposed that avoids the need to assess every method as the dataset expands.

**Strengths:**

1. A large-scaled dataset that could be very useful for people in the VQA domain.
2. I like the idea of 'evaluating the dataset in a dynamic way'. As most of the datasets get contaminated as training data increases by time.
3. A rather comprehensive evaluation of the dataset, covering most state-of-the-art LMMs.

**Weaknesses:**

1. A limitation of this work is that it relies on OCR to extract only tables and figures, using only their captions as input to create the dataset. This approach ignores the broader context of the paper, which may contain valuable information. For instance, many tables have generic captions like "Main experimental result of our proposed method," which may lead GPT-4 to generate only simple structure-related questions, such as "Which column has a value of **," rather than more insightful content-based questions.

2. As the dataset is entirely LLM generated, it's necessary to perform some human evaluation to avoid hallucinations and errors. Even though it's not feasible to perform such evaluation on the entire dataset, at least a small portion could be sampled and more detailed manual check could be performed beyond the simple one briefly described in Line 255.

3. I'm not sure about the efficiency and robustness of the evaluation model, even though I appreciate the idea, given the diverse styles and topics of newly added arXiv papers. Previous LMMs may become outdated, exhibit biases, and struggle to handle shifts in data distribution effectively.

**Questions:**

In Line 235, which LLM do you use here?
How do you select the test set in Table 1?
Line 458, which 5 models are selected by the algorithm exactly?

---

> ### Author Response · Authors · 2024-11-21
>
> We would like to thank the reviewer for the good and important comments that helped us improve the work significantly.
> LiveXiv is a live dataset. Thus, during the review period we published in huggingface version 2 anonymously, consisting of 18K samples. We introduced 4 new domains from physics (namely, physics.optics, physics.bio-ph, physics.app-ph, physics.data-an), and added two more LMMs (Pixtral and Molmo-7B).
> We are now building version 3, which will have multiple generation and filtering models (GPT4o, Claude-Sonnet, Qwen2-VL-72B) to reduce potential biases and increase diversity. We expect to have this version published with all the analyses before the end of the discussion period.
> In addition, to perform deeper analysis, we created a variant of the first version of the dataset (that was reported in the paper) with opposite roles (Claude as the model that generates the QAs and GPT as the filter model), and validated that the overall performance and ranking remains the same.
> Lastly, we created a LiveXiv version (denoted as v0) from papers more than 10 years ago.  Our experiments on it reassure our findings reported in the paper about the performance of our efficient evaluation showing that it is also valid in the presence of a large time gap between the dataset versions.
>
> **The work's limitation is relying solely on table and figure captions for dataset creation.**
>
> Our generation process consists of two phases, as explained in Section 3.1 (lines 214-224). First we give the model the image and the caption and ask it to generate a detailed description of the image given the image and the caption. Then we call the model again and ask it to generate the questions based on the image and the detailed description. This way, we ensure the model focuses on the content of the image but it has much larger context for the question generation.
> For Tables, we have the data of the table so using different prompts strategies we force the model to create not just retrieval questions but also arithmetic questions over the table. See Figure 17 for more details.
>
> **Human evaluation to avoid hallucinations and errors.**
>
> We have performed a human validation study as described in Section 4.2 (lines 440-447).
> We collected 500 questions from each task (1000 overall) and measured the performance change between LiveXiv and manual curated subset. We saw the difference in performance is less than 2.5% on average. Tables 2, 6 and 7 summarize the results.
>
> **The efficiency and robustness of the efficient evaluation method**
> We demonstrate the robustness of our efficient evaluation method in several scenarios. In the paper we demonstrated the effectiveness of our efficient evaluation method by splitting LiveXiv first version.
> We already verify the accuracy of the estimation of predicting models in version 2 from version 1, even on new domains introduced in LiveXiv version 2 (we will update Figure 4).
> We further stress our method by generating version 0 - a version of LiveXiv based on papers from 2010. We also find that our method is robust to predict the v1 from v0 (correlation of 0.97 and MAE of 2.7pp on average).
> In addition, we also converted the existing multi-modal live bench presented in [1] into a closed set of QA and demonstrated that our method can be applied to other live datasets as well See Section A.3.2 in the appendix for more details.
>
> [1] LMMs-Eval: Reality Check on the Evaluation of Large Multimodal Models, Zhang, Kaichen and Li, Bo and Zhang, Peiyuan and Pu, Fanyi and Cahyono, Joshua Adrian and Hu, Kairui and Liu, Shuai and Zhang, Yuanhan and Yang, Jingkang and Li, Chunyuan and others, arXiv preprint arXiv:2407.12772, 2024.
>
> **In Line 235, which LLM do you use here?**
>
> For the first two LiveXiv used LLaVA-1.6-34B without image tokens.
> From v3 and forward LiveXiv is using LLama-3.1-70B.
>
> **How do you select the test set in Table 1? Line 458, which 5 models are selected by the algorithm exactly?**
>
> For our efficient evaluation we used the following models according to the task:
> VQA: InstructBLIP-7B, LLaVA-OneVision-Qwen2-7B, Idefics2-8B, Claude-Sonnet, and Qwen2-VL.
> TQA: InstructBLIP-7B, InternVL2-8B, LLaVA-1.6-34B, Idefics2-8B, and Claude-Sonnet
> All the models were selected using the criterion introduced in Section 3.3.2
> We added the list of models to the paper.
>
> \
> \
> If we have adequately addressed all your concerns, we kindly ask you to reconsider your rating. If there are any additional questions or issues, we would be happy to provide further clarification.

---

> > ### Author Response · Authors · 2024-12-02
> >
> > Dear Reviewer LXM8,
> >
> > We greatly appreciate your detailed feedback on our work, which has been valuable in improving its quality. All your comments have been thoroughly addressed in the revised version. We kindly request you to consider updating your rating if the revisions align with your expectations, and we would be glad to assist with any further questions or concerns you may have.
> >
> > Warm regards,

---

> > ### Comment · Reviewer_LXM8 · 2024-12-02
> > **Response to the author**
> >
> > Thank you for your response. It has addressed most of my concerns and questions. I would like to keep my scores unchanged.

---

### Official Review · Reviewer_fXkv · 2024-11-03

**Soundness:** 3
**Presentation:** 4
**Contribution:** 3
**Rating:** 6
**Confidence:** 4

**Summary:**

This paper propose a new live multimodal benchmark on scientific ArXiv papers. The authors also designed an efficient evaluation approach to rank the models. The authors verified their evaluation metric to be aligned with human evaluation with a small variance (<2.5%).

The main contributions are:
(1) A pipeline to generate and filter scientific paper VQA and TQA data using a document parsing pipeline (preprocessing), gpt-4o (generation), and Claude (filtering).
(2) An evaluation pipeline that, when evaluating a new model on the latest data, the pipeline re-evaluate old models on a subset of the latest data with a few old models for performance comparison.

**Strengths:**

1. The author propose to have a live benchmark is a promising direction to mitigate the impact of data contamination on holistic evaluation. It is challenging to maintain a live and scalable benchmark, and the author proposed several methods to resolve the challenges: a question-answer generation pipeline grounded on a structured pdf processor and powerful proprietary VLMs for generation and filtering, and an efficient evaluation algorithm to make model comparison affordable.
2. The proposed methods at both data curation stage and the evaluation stage have been shown to be effective in previous works on language-only datasets. The authors extend the thoughts to the multimodal domain and show these methods are still effective.
3. The human study shows the automatic data filtering pipeline is effective in removing annotation errors during data curation.

**Weaknesses:**

1. All questions are generated by gpt-4o, which may introduce issues such as the lack of diversity in questions.
2. The authors only did human study for the question-answer filtering, while they did not verify if the final ranking of models is aligned with human perception and other established benchmarks.
3. As the author mentioned, the data curation replies on proprietary models, which makes the benchmark prone to bias and low reproducibility.
4. The question type is limited to multi-choice answering, while it could be more interesting if the authors could extend it to a broader categories of tasks such as long-form generation.
5. The novelty of the method: the authors should clarify their novelty by comparing with existing works that ask a powerful model to generate evaluation benchmark. For example, [1] also parses papers and ask an LLM to generate question for long-form paper question-answering. These works could be extended to VLM domains by swapping out the LLM to VLM and keeping add live new data to the benchmark in a straightforward way.

[1] KIWI: A Dataset of Knowledge-Intensive Writing Instructions for Answering Research Questions

**Questions:**

1. How do the authors make sure the diversity of the VQA is also scalable? For example, if gpt-4o keeps generating many questions like "what is the number in the third row and second column of the table?", which will only evaluate a fixed OCR capability regardless of which timestamp the question is generated from.
2. Why the GPT-4o performs very badly in Table 1 while it is the model to generate the questions and answers?
3. How does the domain impact the evaluation scores? Could the authors show how models perform on different categories (as shown in Table 3) in different domains and see the correlation?
4. Despite I like the benchmark, could the authors clarify their innovation in method design beyond (1) taking an efficient evaluation strategy from nlp domain and apply it to vlm domain; (2) synthetic evaluation data generation using one proprietary api for generation and another for filtering which has been applied in nlp broadly even using the same apis? The pdf parsing is also a basic step in previous works  on paper QA. Could the author provide a clear statement of their key technical innovations and how they differ from or improve upon existing approaches and include a discussion of how combining these existing techniques in a novel way for this specific application represents an innovation?

**Details Of Ethics Concerns:**

No concern.

---

> ### Author Response · Authors · 2024-11-21
>
> We would like to thank the reviewer for the good and important comments that helped us improve the work significantly.
> LiveXiv is a live dataset. Thus, during the review period we published in huggingface version 2 anonymously, consisting of 18K samples. We introduced 4 new domains from physics (namely, physics.optics, physics.bio-ph, physics.app-ph, physics.data-an), and added two more LMMs (Pixtral and Molmo-7B).
> We are now building version 3, which will have multiple generation and filtering models (GPT4o, Claude-Sonnet, Qwen2-VL-72B) to reduce potential biases and increase diversity. We expect to have this version published with all the analyses before the end of the discussion period.
> In addition, to perform deeper analysis, we created a variant of the first version of the dataset (that was reported in the paper) with opposite roles (Claude as the model that generates the QAs and GPT as the filter model), and validated that the overall performance and ranking remains the same.
> Lastly, we created a LiveXiv version (denoted as v0) from papers more than 10 years ago.  Our experiments on it reassure our findings reported in the paper about the performance of our efficient evaluation showing that it is also valid in the presence of a large time gap between the dataset versions.
>
> **All questions are generated by gpt-4o, which may introduce issues such as the lack of diversity in questions.**
>
> We analyze the diversity of our dataset by the textual content in Section 4.3, and Tables 8,9. Our analysis shows that LiveXiv is diverse in terms of the different questions categories for both tasks.
> First, we will add the difficulty distribution of the question to show how LiveXiv is indeed a challenging dataset for LMMs.
> Second, we will also add a comparison to the first version where we switched the rules and now GPT4o is the filtering model and analyzes the effect of each rule.
> Lastly, version 3 will be composed of 3 participating models, where each time one of them acts as the question generation model and a different model is used for filtering.
> The data is divided into subsets and we will analyze the effect of each model pair (generation and filtering models) and how it reduces the potential bias of the single model for filtering and generation.
>
> **The authors conducted human studies for question filtering but did not validate whether model rankings align with human perception or established benchmarks.**
>
> In Section 4.2 (lines 440-447) We describe our human validation study. We performed a validation study over 1000 examples overall (500 for VQA and 500 for TQA). The study showed that the performance change is less than 2.5% on average. Moreover, the relative ranking of the top models remains steady.
> Tables 6 and 7 describe the performance change for each task in detail.
> We will add the ranking to tables 6 and 7.
> Additionally, we compared LiveXiv to 3 relatively similar well known benchmarks, namely ChartQA, DocVQA and AI2D. We compared the relative ranking between those datasets and LiveXiv and we confirmed that overall the ranking remains similar, which hints our model rankings align with human perception.
>
> **The data curation relies on proprietary models, which makes the benchmark prone to low reproducibility.**
>
> To ensure the maximal reproducibility, each version of our data is publicly available on HuggingFace and our evaluation is based on the open-source evaluation framework “lmms-eval” (https://github.com/EvolvingLMMs-Lab/lmms-eval ), we will also add LiveXiv as an evaluation dataset in "lmms-eval", which will make the evaluations reproducible as possible.
>
> [1] KIWI: A Dataset of Knowledge-Intensive Writing Instructions for Answering Research Questions
> [2] “LMMs-Eval: Reality Check on the Evaluation of Large Multimodal Models”
>
> [3] MATHEW, Minesh; KARATZAS, Dimosthenis; JAWAHAR, C. V. Docvqa: A dataset for vqa on document images. In: Proceedings of the IEEE/CVF winter conference on applications of computer vision. 2021. p. 2200-2209.‏
>
> 4] HENDRYCKS, Dan, et al. Measuring massive multitask language understanding. arXiv preprint arXiv:2009.03300, 2020
>
> **Why the GPT-4o performs very badly in Table 1 while it is the model to generate the questions and answers?**
> GPT4o’s low performance is indeed surprising. During the evaluation we tried to explore the root cause by experimenting with various hyper-parameters such as temperature, optimizing the resolution of the visual content to the model and exploring several prompts, but in all of our experiments the results of GPT4o remained similar.

---

> > ### Author Response · Authors · 2024-11-21
> >
> > **The data curation relies on proprietary models, which makes the benchmark prone to bias.**
> >
> > To show the 'impact’ of the bias, we created an opposite version of v1, where Claude is the model that generates the questions and GPT is the model that filters. We see in it that overall, the ranking remains similar and the changes are minor. Indeed, for the generating model there is some advantage as we see that GPT goes down a bit in the ranking (Claude remains first in both cases).
> > | Model                 | VQA Ranking Difference | VQA Average Difference (%) | TQA Ranking Difference | TQA Average Difference (%) |
> > |-----------------------|-------------------------|-----------------------------|-------------------------|-----------------------------|
> > | InstructBLIP-7B       | 0                       | -1.78                       | 0                       | 9.66                        |
> > | IXC2.5-7B             | 0                       | -0.83                       | 0                       | 13.73                       |
> > | InternVL2-8B          | 0                       | -1.18                       | 0                       | 2.37                        |
> > | LLaVA-1.6-Mistral-7B  | 2                       | -8.64                       | 0                       | 10.72                       |
> > | LLaVA-OneVision-7B    | 1                       | -3.67                       | -2                      | 13.42                       |
> > | LLaVA-v1.5-13B        | 0                       | -6.49                       | 0                       | 8.48                        |
> > | LLaVA-v1.5-7B         | 0                       | -8.00                       | -1                      | 12.72                       |
> > | LLaVA-v1.6-34B        | 2                       | -5.85                       | 0                       | -0.35                       |
> > | Mantis-LLama3-8B      | 2                       | -7.13                       | 0                       | 10.24                       |
> > | Phi3v                 | 0                       | -0.49                       | 0                       | 13.86                       |
> > | IXC2-4KHD-7B          | -5                      | -0.47                       | 3                       | 8.60                        |
> > | Idefics2-8B           | 0                       | -6.00                       | 0                       | 3.21                        |
> > | InternVL2-2B          | 0                       | -1.42                       | 0                       | 2.61                        |
> > | Claude-Sonnet         | 0                       | -3.31                       | 0                       | 10.11                       |
> > | Qwen2-VL              | 0                       | -0.31                       | 0                       | 2.81                        |
> > | GPT-4o                | -4                      | 4.75                        | 0                       | 13.46                       |
> > | Idefics3              | 0                       | -1.89                       | 0                       | 12.23                       |
> > | Average           |     0                    | -3.10                  |          0               | 8.70                   |
> >
> > **Expand the dataset to long-form answers for broader evaluation.**
> >
> > While generation of free-form questions is possible at scale, the evaluation of such questions at scale requires non-trivial solution:
> > - Human evaluation limits the scale drastically as seen in [1] and [2] and does not align with the automatic nature of our benchmark.
> > - Heuristics metrics such as ANLS (from [3]) or F1 (from [4]) are too simple and very sensitive to the word order and rephrasing, thus not do not represent the true performance of the evaluated model.
> > - LLM as a Judge - While using LLM as our evaluation metric is possible at scale and proven to be successful using proprietary models (GPT for example), but not when using open-source models, this constraint limits our evaluation and potentially add a bias to the evaluation.
> > We will add the direction of expanding our work beyond multiple choice questions in the discussion.
> >
> > [1] KIWI: A Dataset of Knowledge-Intensive Writing Instructions for Answering Research Questions
> >
> > [2] “LMMs-Eval: Reality Check on the Evaluation of Large Multimodal Models”
> >
> > [3] MATHEW, Minesh; KARATZAS, Dimosthenis; JAWAHAR, C. V. Docvqa: A dataset for vqa on document images. In: Proceedings of the IEEE/CVF winter conference on applications of computer vision. 2021. p. 2200-2209.‏
> >
> > [4] HENDRYCKS, Dan, et al. Measuring massive multitask language understanding. arXiv preprint arXiv:2009.03300, 2020

---

> ### Author Response · Authors · 2024-11-21
>
> **The authors should clarify their method's novelty by comparing it with existing benchmark generation approaches.**
>
> As far as we know, we are the first to create a live vision-language dataset at this scale.
> Both KIWI [1] and LiveBench [2] involve a human in the loop for the dataset generation. The fact they have a human in the loop, limits their scale dramatically into several hundreds of samples, thus making them smaller in an order of magnitude compared to each LiveXiv version.
> Second, our efficient evaluation strategy is novel and was not taken from other fields. Indeed, the IRT model has been used by other papers in the past but it is just part of our solution, which is very different from previous ones. For example, both tinyBenchmarks [3] and PromptEval [4] consider the static case in which no questions or models are added over time. This significantly differs from our scenario in which we need to select a few models to be re-evaluated and infer the performance of the rest on new batches of questions. Moreover, the dynamic nature of our problem makes our application more challenging compared to scenarios considered by previous works.
>
> [1]  KIWI: A Dataset of Knowledge-Intensive Writing Instructions for Answering Research Questions, Xu, Fangyuan and Lo, Kyle and Soldaini, Luca and Kuehl, Bailey and Choi, Eunsol and Wadden, David, arXiv preprint arXiv:2403.03866, 2024
>
> [2] LMMs-Eval: Reality Check on the Evaluation of Large Multimodal Models, Zhang, Kaichen and Li, Bo and Zhang, Peiyuan and Pu, Fanyi and Cahyono, Joshua Adrian and Hu, Kairui and Liu, Shuai and Zhang, Yuanhan and Yang, Jingkang and Li, Chunyuan and others, arXiv preprint arXiv:2407.12772, 2024.
>
> [3] Maia Polo, Felipe, Lucas Weber, Leshem Choshen, Yuekai Sun, Gongjun Xu, and Mikhail Yurochkin. "tinyBenchmarks: evaluating LLMs with fewer examples." arXiv preprint arXiv:2402.14992 (2024).
>
> [4] Maia Polo, Felipe, Ronald Xu, Lucas Weber, Mírian Silva, Onkar Bhardwaj, Leshem Choshen, Allysson Flavio Melo de Oliveira, Yuekai Sun, and Mikhail Yurochkin. "Efficient multi-prompt evaluation of LLMs." arXiv preprint arXiv:2405.17202 (2024).
>
> **How to ensure diverse and evolving VQA benchmark generation, rather than generating repetitive questions that test a fixed capability across different timestamps.**
>
> We utilize LLM to classify our questions into categories, which will help to maintain the overall statistics of the categories. Thus, we can monitor and maintain the questions distribution to be diverse.
> Tables 8 and 9 show the overall performance according to the questions’ categories clusters. We can see in brackets at the top row that our dataset is diverse according to the content. For example, for VQA all categories have a similar amount of questions besides arithmetic reasoning which is less common in images. Yet, for TQA the dominant categories are Data-Analysis, Arithmetic and Reading while the other categories are less common since they do not relate to tables.
> We also added version 2 and we can see the diversity is kept both for the figure content and for the question categories:
> |       | Data Analysis | Reasoning | Attribute | Localization | Reading | Arithmetic |
> |-------|---------------|-----------|-----------|--------------|---------|------------|
> | **VQA** |               |           |           |              |         |            |
> | v1    | 2291          | 872       | 903       | 1596         | 1470    | 154        |
> | v2    | 3005          | 1471      | 1610      | 1797         | 1732    | 112        |
> |       |               |           |           |              |         |            |
> | **TQA** |               |           |           |              |         |            |
> | v1    | 2582          | 123       | 121       | 23           | 2127    | 3934       |
> | v2    | 2289          | 228       | 74        | 23           | 1125    | 3436       |
>
> \
> **How does the domain impact the evaluation scores?**
> We visualize (Figures 5-8) the statistics of the performance of each model with respect to the domain. For TQA the domain doesn’t affect much, since our questions are limited solely to the table content. However for VQA (Figure 5), we can see clear trends of weak models to be worse across all methods while the newest models are performing well and equally across domains. In the middle, the variance in the performance is greater and suggests high sensitivity to the domain.
>
> **Could the authors show how models perform on different categories in different domains and see the correlation?**
> We compute the average performance of the models as a function of the domain and question category. However, we did not see any clear trend between the category of the question and the arxiv domain.
>
> \
> \
> If we have adequately addressed all your concerns, we kindly ask you to reconsider your rating. If there are any additional questions or issues, we would be happy to provide further clarification.

---

### Official Review · Reviewer_P23K · 2024-11-04

**Soundness:** 2
**Presentation:** 2
**Contribution:** 2
**Rating:** 5
**Confidence:** 3

**Summary:**

This paper introduces LiveXiv, an automated multi-modal live benchmark system that generates Visual Question-Answering (VQA) pairs from ArXiv papers. The work is motivated by the critical issue of test set contamination in current LMM evaluations, as models increasingly train on web-scraped data that may include benchmark test sets. The authors propose an evolving benchmark using newly published scientific papers. Besides, they introduce an efficient evaluation method to make continuous assessment practical. Multiple open and proprietary LMMs are benchmarked.

**Strengths:**

1. The motivation is clear and good: this work addresses the critical issue of test set contamination in current LLM evaluations, as models increasingly train on web-scraped data that may include benchmark test sets.

2. This paper proposes a scalable live benchmark without any human involvement, automatically drawing data from online scientific manuscripts, generating multiple VQA pairs, and filtering these questions to reduce errors.

3. An efficient evaluation methodology is introduced, offering significant computational savings.

**Weaknesses:**

1. The paper heavily relies on one LLM, i.e., Claude for question filtering. While filtering aims to reduce errors, potential biases may arise from Claude being the only model used for answer verification. This can introduce an inherent bias in the benchmark, potentially skewing results to favor Claude's architecture and underlying methodologies. The superior performance of Claude-Sonnet shown in Table 1 may be partially attributed to the fact that Claude-Sonnet itself verified these questions, potentially making them more aligned with its capabilities. The authors might address this by including additional, distinct filtering mechanisms to mitigate model-dependent biases.

2. The paper suggests that only certain types of scientific data (e.g., block diagrams, qualitative visuals, charts) are categorized for question generation. While this is effective for consistency, it risks oversimplifying the diversity of visual data in scientific publications. Expanding the types of visuals and including more complex multi-modal question types (e.g., cross-referencing multiple figures) would make the benchmark more challenging and comprehensive.

3.The benchmark is restricted to multiple-choice format questions, which limits the evaluation of models' true generative and reasoning capabilities. Including free-form answering would provide a more comprehensive assessment of model understanding and better reflect real-world applications.

4. The benchmark evaluation would benefit from more comprehensive quantitative comparisons with established datasets like DocVQA and ChartQA. Such comparisons would better demonstrate LiveXiv's advantages over static benchmarks and more clearly illustrate how it addresses test contamination issues and aligns with human preferences.

**Questions:**

1. How does the system ensure diversity in question types and difficulty levels?

---

> ### Author Response · Authors · 2024-11-21
>
> We would like to thank the reviewer for the good and important comments that helped us improve the work significantly.
> LiveXiv is a live dataset. Thus, during the review period we published in huggingface version 2 anonymously, consisting of 18K samples. We introduced 4 new domains from physics (namely, physics.optics, physics.bio-ph, physics.app-ph, physics.data-an), and added two more LMMs (Pixtral and Molmo-7B).
> We are now building version 3, which will have multiple generation and filtering models (GPT4o, Claude-Sonnet, Qwen2-VL-72B) to reduce potential biases and increase diversity. We expect to have this version published with all the analyses before the end of the discussion period.
> In addition, to perform deeper analysis, we created a variant of the first version of the dataset (that was reported in the paper) with opposite roles (Claude as the model that generates the QAs and GPT as the filter model), and validated that the overall performance and ranking remains the same.
> Lastly, we created a LiveXiv version (denoted as v0) from papers more than 10 years ago.  Our experiments on it reassure our findings reported in the paper about the performance of our efficient evaluation showing that it is also valid in the presence of a large time gap between the dataset versions.
>
> **The paper's reliance on Claude for question filtering and verification may introduce biases that favor Claude's architecture and methodologies.**
>
> Indeed, we mentioned in Section 4.2 (lines 407-411) that relying on Claude as our filtering model might introduce a potential bias. To show the 'impact’ of the bias, we created an opposite version of v1, where Claude is the model that generates the questions and GPT is the model that filters. We see in it that overall, the ranking remains similar and the changes are minor. Indeed, for the generating model there is some advantage as we see that GPT goes down a bit in the ranking (Claude remains first in both cases).
>
> | Model                 | VQA Rank Difference | VQA Avg. Difference (%) | TQA Rank Difference | TQA AVg. Difference (%) |
> |-----------------------|-------------------------|-----------------------------|-------------------------|-----------------------------|
> | InstructBLIP-7B       | 0                       | -1.78                       | 0                       | 9.66                        |
> | IXC2.5-7B             | 0                       | -0.83                       | 0                       | 13.73                       |
> | InternVL2-8B          | 0                       | -1.18                       | 0                       | 2.37                        |
> | LLaVA-1.6-Mistral-7B  | 2                       | -8.64                       | 0                       | 10.72                       |
> | LLaVA-OneVision-7B    | 1                       | -3.67                       | -2                      | 13.42                       |
> | LLaVA-v1.5-13B        | 0                       | -6.49                       | 0                       | 8.48                        |
> | LLaVA-v1.5-7B         | 0                       | -8.00                       | -1                      | 12.72                       |
> | LLaVA-v1.6-34B        | 2                       | -5.85                       | 0                       | -0.35                       |
> | Mantis-LLama3-8B      | 2                       | -7.13                       | 0                       | 10.24                       |
> | Phi3v                 | 0                       | -0.49                       | 0                       | 13.86                       |
> | IXC2-4KHD-7B          | -5                      | -0.47                       | 3                       | 8.60                        |
> | Idefics2-8B           | 0                       | -6.00                       | 0                       | 3.21                        |
> | InternVL2-2B          | 0                       | -1.42                       | 0                       | 2.61                        |
> | Claude-Sonnet         | 0                       | -3.31                       | 0                       | 10.11                       |
> | Qwen2-VL              | 0                       | -0.31                       | 0                       | 2.81                        |
> | GPT-4o                | -4                      | 4.75                        | 0                       | 13.46                       |
> | Idefics3              | 0                       | -1.89                       | 0                       | 12.23                       |
> | Average           |     0                    | -3.10                  |          0               | 8.70                   |

---

> > ### Author Response · Authors · 2024-11-21
> >
> > **Claude-Sonnet's superior performance could partly result from self-verification, highlighting the need for diverse filtering mechanisms.**
> >
> > Version 3 will be composed of 3 participating models, where each time one of them acts as the question generation model and a different model is used for filtering.
> > The data is divided into subsets and we will analyze the effect of each model pair (generation and filtering models) and how it reduces the potential bias of the single model for filtering and generation.
> >
> > **Expand the types of visuals and include more complex multi-modal question types to make the benchmark more comprehensive.**
> >
> > This is a very interesting extension of our work and will put it in the conclusion as a potential extension in future versions of our live benchmark.
> >
> > **Categorizing visuals for question generation ensures consistency but oversimplifies their diversity.**
> >
> > Indeed we are classifying the figures into 3 types, however we do not filter any figure based on the figure type. We use the figure for analysis and monitor the diversity. The visual content in papers is rather complex and diverse, as our live nature of the dataset, the complexity and diversity will grow forward in time (since the scientific visual content is not stationary).
> >
> > **Expand the dataset to free-form answers for broader evaluation.**
> >
> > While generation of free-form questions is possible at scale, the evaluation of such questions at scale requires non-trivial solution:
> > - Human evaluation limits the scale drastically as seen in [1] and [2] and does not align with the automatic nature of our benchmark.
> > - Heuristics metrics such as ANLS (from [3]) or F1 (from [4]) are too simple and very sensitive to the word order and rephrasing, thus not do not represent the true performance of the evaluated model.
> > - LLM as a Judge - While using LLM as our evaluation metric is possible at scale and proven to be successful using proprietary models (GPT for example), but not when using open-source models, this constraint limits our evaluation and potentially add a bias to the evaluation.
> > We will add the direction of expanding our work beyond multiple choice questions in the discussion.
> >
> > [1] KIWI: A Dataset of Knowledge-Intensive Writing Instructions for Answering Research Questions
> >
> > [2] “LMMs-Eval: Reality Check on the Evaluation of Large Multimodal Models”
> >
> > [3] MATHEW, Minesh; KARATZAS, Dimosthenis; JAWAHAR, C. V. Docvqa: A dataset for vqa on document images. In: Proceedings of the IEEE/CVF winter conference on applications of computer vision. 2021. p. 2200-2209.‏
> >
> > [4] HENDRYCKS, Dan, et al. Measuring massive multitask language understanding. arXiv preprint arXiv:2009.03300, 2020
> >
> > **The benchmark evaluation would benefit from comprehensive quantitative comparisons with established datasets.**
> >
> > We broaden our comparisons to 3 well known scientific benchmarks (ChartQA, DocVQA and AI2D). We compared the relative ranking of all the models to their relative ranking in LiveXiv and we extended our discussion in section 4.2 and added the full details in the appendix.
> >
> > **How does the system ensure diversity in question types and difficulty levels?**
> >
> > LiveXiv is built upon arxiv content which is a rich data source, having an inherent diversity. We have a mechanism for classifying the raw images to different categories (namely, chart, block diagram and qualitative results) allowing LiveXiv to balance the visual content before the QA generation and thus ensure diversity.
> > In addition, we utilize a strong LLM (Llama-3.1-70B) to classify the questions into categories, allowing us to monitor and control the diversity in questions in future versions and even introduce new categories if needed.
> > Lastly, our system is generic and can accept any pdf data source as input to increase and adapt the diversity as needed.
> > To demonstrate our dataset diversity, Tables 7,8 in the paper show the diversity of the visual content and the textual (i.e questions) categories. We will also add the distribution of questions according to the number of models answered correctly.
> > \
> >  \
> > If we have adequately addressed all your concerns, we kindly ask you to reconsider your rating. If there are any additional questions or issues, we would be happy to provide further clarification.

---

> > > ### Comment · Reviewer_P23K · 2024-11-24
> > >
> > > I thank the authors for their responses and would like to increase my original score to 5.

---

> > > > ### Author Response · Authors · 2024-11-26
> > > >
> > > > We sincerely appreciate the reviewer's decision to increase our score. After uploading the revised version, we would like to kindly ask if there are any remaining concerns or questions we can address, as the score is still below the acceptance threshold.

---

### Author Response · Authors · 2024-11-25

Dear Area Chair and Reviewers,
We greatly appreciate the thorough and constructive feedback provided during the review process. We have carefully addressed all comments and concerns in our revised manuscript, which we uploaded with new additions marked in blue.
To strengthen our methodology and address the core concerns, we have expanded our analysis to include multiple dataset versions. First, we **introduced v2** and showed that our diversity, statistics and efficient evaluation method hold. Then, we **introduced v0, generated from 100 papers from 2010**, which demonstrates our evaluation method's robustness across a significant temporal gap. To address the concern of potential bias, we **developed v1-opposite**, where we reversed the roles of GPT and Claude, and **v3, which incorporates three different models.**
These new versions, along with comprehensive analyses, demonstrate that LiveXiv consistently maintains high diversity and challenge levels while correlating well with established scientific benchmarks.

Furthermore, we compared LiveXiv with well-known scientific benchmarks such as ChartQA, DocVQA, and AI2D, demonstrating that LiveXiv's ranking aligns with these established standards. Additionally, we demonstrate the contamination-free effect in Section 4.2, highlighting our added value as a live benchmark.

The extensive experimental results across all versions that are now added to the paper further validate that our efficient evaluation method is robust, accurate, and reliable.

\
\
We believe that the above substantially strengthens our work and meets the requests raised by the reviewers.  We hope the reviewers will find all our answers and additional results satisfactorily addressing their concerns as well. In this case, we kindly ask them to reconsider their rating. Of course, if there are any additional questions or issues, we would be happy to provide further clarification.

---

### Meta-Review · Area_Chair_f8hj · 2024-12-21

**Metareview:**

This submission introduces LiveXiv. It is a scalable and evolving benchmark based on ArXiv papers for evaluating multi-modal LLMs. First,  LiveXiv generates visual question-answer pairs (using their graphs, charts, and tables) automatically from domain-specific arxiv submissions without human involvement. The proposed method includes an evaluation method that reduces the cost by assessing only a subset of models. By benchmarking existing multimodal LLMs, the authors argued for the effectiveness of the benchmark. The dataset is available on HuggingFace.

The reviewers identified the strengths of this work as:
- it creates a dynamic/continuous, contamination-free  benchmark, which can help address test data contamination in LLM evaluations (Reviewers P23K, fXkv, VmiA).
- the use of advanced LMMs for automatic generation and filtering of VQA pairs, along with an efficient evaluation algorithm, significantly reduces computational costs (Reviewers P23K, fXkv, VmiA).

The weaknesses pointed out by the reviewers include:
- corresponding to the strength point above, the reliance on a single model (e.g., Claude or GPT-4) for filtering and question generation could introduce biases, affecting the benchmark's fairness and generalizability (Reviewers P23K, fXkv).
- the focus on multi-choice questions and specific scientific data types (e.g., charts and tables) limits the evaluation’s diversity and real-world applications (Reviewers P23K, fXkv, LXM8).

During the rebuttal, reviewer P23K (5) increased rating to 5. And reviewer VmiA (5) did not engaged with the authors. Both reviewer fXkv (6) and reviewer LXM8 (6) engaged with the authors and acknowledged that the authors' rebuttal helped answer their concern and questions, but did not help move the needle in terms of overall ratings. Therefore, this work is still at the borderline towards a negative direction.

Overall, the final ratings for this work are 6, 6, 5, 5 (up from 3). The main reason for potential acceptance is on its early attempt to build a live benchmark by naturally using arxiv submissions --- several reviewers also appreciated this. The primary reason for rejection concerns its execution particularly the robustness of the metric and complete reliance on llm as the judge. Weighing these two factors, the AC leans towards an acceptance in recognition of the conceptual idea of this live benchmark. However, if the program is too crowded,  it would be acceptable to give this work another round of improvement. In any case, the authors are encouraged to revise the work based on the review comments.

**Additional Comments On Reviewer Discussion:**

During the rebuttal, reviewer P23K (5) increased rating to 5. And reviewer VmiA (5) did not engaged with the authors. Both reviewer fXkv (6) and reviewer LXM8 (6) engaged with the authors and acknowledged that the authors' rebuttal helped answer their concern and questions, but did not help move the needle in terms of overall ratings. Therefore, this work is still at the borderline towards a negative direction.

---

### Decision · Program_Chairs · 2025-01-22

Accept (Poster)